# Long-term validation of Aeolus L2B wind products at Punta Arenas, Chile and Leipzig, Germany

Holger Baars[1], Joshua Walchester[1,3], Elizaveta Basharova[1,3], Henriette Gebauer[1,3], Martin Radenz[1], Johannes Bühl[1], Boris Barja[2], Ulla Wandinger[1], and Patric Seifert[1]

[1]Leibniz Institute for Tropospheric Research (TROPOS), Leipzig, Germany
[2]Atmospheric Research Laboratory, University of Magallanes, Punta Arenas, Chile
[3]University of Leipzig, Leipzig, Germany

**Correspondence:** Holger Baars, baars@tropos.de

**Abstract.** Ground-based observations of horizontal winds have been performed at Leipzig (51.35° N, 12.43° E), Germany, and at Punta Arenas (53.15° S, 70.91° W), Chile, in the framework of the German initiative EVAA (Experimental Validation and Assimilation of Aeolus observations) with respect to the validation of the Mie and Rayleigh wind products of Aeolus (L2B data). In Leipzig, at the Leibniz Institute for Tropospheric Research (TROPOS), radiosondes have been launched for the Aeolus overpasses on each Friday (ascending orbit) since mid of May 2019. In Punta Arenas, scanning Doppler cloud radar observations have been performed in the framework of the DACAPO-PESO campaign (dacapo.tropos.de) for more than 3 years from the end of 2018 until the end of 2021 and could be used to validate Aeolus measurements on its ascending and descending orbit. We present two case studies and long-term statistics of the horizontal winds derived with the ground-based reference instruments compared to Aeolus Horizontal Line-of-Sight (HLOS) winds. The wind products of Aeolus considered are the Mie cloudy and Rayleigh clear products. It was found that the deviation of the Aeolus HLOS winds from the ground reference is usually of Gaussian shape, which allowed the use of the median bias and the scaled median absolute deviation (MAD) for the determination of the systematic and random errors of Aeolus wind products, respectively. The case study from August 2020 with impressive atmospheric conditions at Punta Arenas shows that Aeolus is able to capture strong wind speeds up to more than $100\,\mathrm{ms^{-1}}$.

The long-term validation has been performed in Punta Arenas covering the period from December 2018 to November 2021 and in Leipzig from May 2019 until September 2022. This analysis showed that the systematic error of the Aeolus wind products could be significantly lowered during the mission lifetime with the changes introduced into the processing chain (different versions are called baselines). While in the early mission phase, systematic errors of more than $2\,\mathrm{ms^{-1}}$ (absolute values) were observed for both wind types (Mie and Rayleigh), these biases could be reduced with the algorithm improvements, such as the introduction of the correction for temperature fluctuations at the main telescope of Aeolus (M1 temperature correction) with Baseline 09. Hence, since Baseline 10, a significant improvement of the Aeolus data was found leading to a low systematic error (close to $0\,\mathrm{ms^{-1}}$) and nearly similar values for the mid-latitudinal sites on both hemispheres. The random errors for both wind products were first decreasing with increasing baseline but later increasing again due to performance losses of the Aeolus lidar instrument. Nevertheless, no significant increase in the systematic error of the Aeolus wind products was found. Thus, one can conclude that the uncertainty introduced by the reduced atmospheric return signal received by Aeolus is mostly affecting

the random error.

Even when considering all the challenges during the mission, we can confirm the general validity of Aeolus observations during its lifetime. Therefore, this space explorer mission could demonstrate that it is possible to perform active wind observations from space with the applied technique.

# 1  Introduction

In 2018, the Aeolus satellite of the European Space Agency (ESA) was launched with the goal to improve weather forecast through global measurements of wind profiles (Stoffelen et al., 2005; Reitebuch, 2012). To obtain vertically resolved wind measurements around the globe, the High-Spectral-Resolution (HSR) Doppler lidar ALADIN (Atmospheric Laser Doppler Instrument) was installed on board. It has been the first time that a lidar with Doppler capabilities as well as with high-spectral-resolution capabilities has been operated in space. Given this unique and novel space-borne technique, it is possible to actively measure vertical profiles of the line-of-sight (LOS) wind in clear sky by using air molecules as tracer (Rayleigh methodology) and in cloudy atmospheric regions by using cloud particles as tracer (Mie methodology, de Kloe et al., 2023; Tan et al., 2008; Baars et al., 2020b). The profiles of LOS wind velocity (measured at 35° off nadir) are then projected to a plane parallel to the Earth's surface to obtain the horizontal line-of-sight (HLOS) wind, i.e., one wind component of the horizontal wind vector (near west-east direction). Besides that, the HSR lidar technology onboard Aeolus can be used to obtain profiles of aerosol and cloud optical properties as spin-off product (e.g., Flament et al., 2021; Baars et al., 2021; Siomos et al., 2021; Abril-Gago et al., 2022a; Gkikas et al., 2023).

The main goal of the mission is the assimilation of the wind products into numerical weather prediction (NWP) models to demonstrate its benefit for weather forecast (Stoffelen et al., 2006; ESA, 2008, 2023). This has meanwhile been done at several meteorological centers (Rennie et al., 2021; Rani et al., 2022; Martin et al., 2022) and a clear positive impact on forecast skills has been reported (ECMWF, 2019a, b).

Given the novelty, extensive validation efforts (Calibration/Validation - Cal/Val) have been needed to verify the observations, detect unforeseen challenges (instrument and processing wise) and develop respective correction or calibration updates in order to make such a data assimilation within near-real time (less than 3 hours) possible at all. For this reason, an intense feedback from Cal/Val teams was desired and obtained to work together with the Aeolus DISC (Data, Innovation, and Science Cluster) and ESA itself on the improvement and stability of instrument and products.

Since the launch in 2018, several challenges were identified by DISC and ESA according to the feedback from the Cal/Val teams (e.g., Krisch et al., 2020). Some important issues are listed in the following:

- Lower laser energy with a more rapid decline than expected (Simonelli et al., 2019; Reitebuch et al., 2020; Lux et al., 2020b),

- Switch to second laser with different beam characteristics which were also changing over time (Straume et al., 2020),

– Occurrence of increased background noise for some pixels (hot pixels) on the ACCD (Accumulation Charge-Coupled Devices) of ALADIN (Weiler et al., 2021a),

– Changes in the wind accuracy according to differences in temperature at the main telescope mirror of ALADIN (Weiler et al., 2021b).

DISC and ESA have worked hard on these features to improve the stability of the instrument and its products which is a prerequisite for the direct assimilation. As the above-mentioned issues influence the use, e.g., the assimilation, of the Aeolus data, continuous and long-lasting validation becomes very important. Most of the operational validation of Aeolus products was performed with NWP models (using of course also assimilated measurements, Chen et al., 2021; Hagelin et al., 2021; Martin et al., 2021; Liu et al., 2022; Zuo et al., 2022), while a direct validation with dedicated measurements has been rare or covered only a short period and usually only a certain geographic region (Baars et al., 2020a; Witschas et al., 2020; Lux et al., 2020a; Baars et al., 2020b; Chen et al., 2021; Martin et al., 2021; Belova et al., 2021; Iwai et al., 2021; Zuo et al., 2022; Wu et al., 2022; Geiß et al., 2022; Lux et al., 2022b; Witschas et al., 2022).

The Leibniz Institute for Tropospheric Research (TROPOS) performed direct, long-term Aeolus-dedicated measurements of the wind vector at two distinct locations in the framework of the cooperation project EVAA (Experimental Validation and Assimilation of Aeolus observations, Baars et al., 2020a) between the Ludwig-Maximilians-Universität of Munich, the German Aerospace Center (DLR), the German Meteorological Service (DWD) and TROPOS. In Leipzig, Germany (51.35° N, 12.43° E), dedicated radiosondes have been launched for the weekly overpass on Aeolus' ascending orbit since May 2019. We analysed this radiosonde data covering the period from June 2019 until September 2022, thus the mission period for which the second laser (FM-B) was in operation. In Punta Arenas, Chile (53.15° S, 70.91° W), continuous remote-sensing observations of LACROS (Radenz et al., 2021) served as one of the very rare Southern Hemispheric Aeolus validation sites (Ratynski et al., 2023; Zuo et al., 2022). Namely, scanning Doppler cloud radar data, which have been collected in the framework of the DACAPO-PESO campaign (Radenz et al., 2021) from November 2018 until November 2021 were used for validation activities covering the ascending and descending orbit. In addition, occasional radiosondes were launched at the local airport, but due to the low amount, we have used this data only for case study validation of both, Aeolus and the ground-based cloud radar observations, but not for the statistical validation approach.

In the work presented here, we assessed the performance of Aeolus and its wind products (both Rayleigh clear and Mie cloudy) for a large part of its lifetime by using our long-term reference measurements. We also evaluated the potential improvements of the wind products by the introduction of new algorithm versions (baselines) into the operational retrieval chain. We thus aimed for analysing the overall performance of Aeolus and the improvements introduced by new processor versions and calibration strategies based on two single sites located in the northern and southern mid-latitudes.

The paper is structured as follows: In Section 2, the experimental setup including the campaign locations and instrumentation is described. In Section 3, we explain the methodology applied to derive horizontal wind from the ground-based remote-sensing instrument and our validation strategy with respect to Aeolus. Selected case studies are discussed in Section 4 to explain the methodology and show already some potentials and limitations of Aeolus. Finally, long-term statistics are presented and dis-

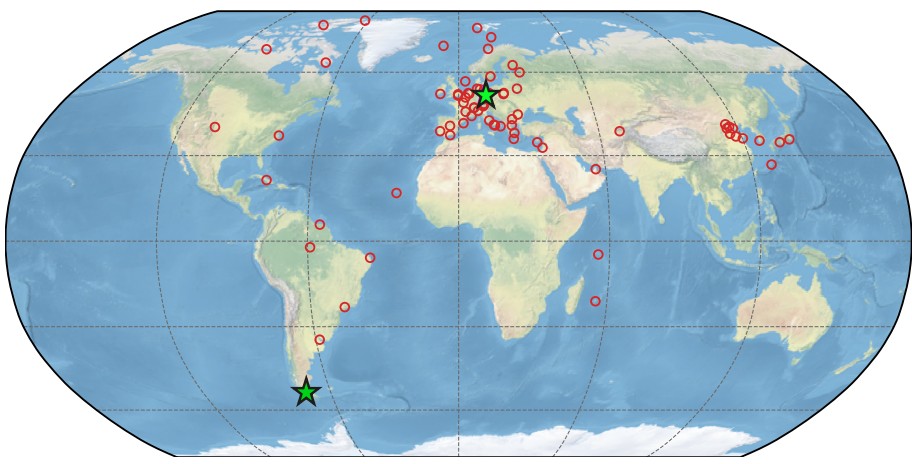

**Figure 1.** Map of Aeolus Cal/Val stations. The ones considered in this work are highlighted as green stars.

cussed with respect to the different algorithm versions and the overall performance of Aeolus during its lifetime in Section 5 and 6. Last but not least, the drawn conclusions are presented.

## 2   Experimental setup

### 2.1   Campaign locations

Measurements at two mid latitudinal locations have been used for the validation activities described here: Leipzig, Germany and Punta Arenas, Chile. Their locations are shown in Fig. 1 together with other ground-based stations which contribute to the validation of Aeolus.

#### 2.1.1   Punta Arenas, Chile

The remote-sensing supersite LACROS (Leipzig Aerosol and Cloud Remote Observations System) has been operated at Punta
Arenas, Chile (53.15° S, 70.91° W) from November 2018 to November 2021 for the DACAPO-PESO campaign (Dynamics, Aerosol, Cloud And Precipitation Observations in the Pristine Environment of the Southern Ocean, Radenz et al., 2021). Thereby, one of the first multi-year ground-based remote sensing data set in the southern mid-latitudes was obtained. The LACROS instrumentation comprises a PollyXT Raman-polarization lidar (Engelmann et al., 2016; Baars et al., 2016), a CHM15kx ceilometer, a MIRA-35 scanning cloud Doppler radar (Görsdorf et al., 2015), a HATPRO microwave radiome-
ter, and a Streamline Doppler lidar. Additionally, radiosondes (Lockheed Martin LMS6) could be launched at the airport of Punta Arenas for dedicated objectives.

Punta Arenas is an ideal location for the validation of Aeolus in terms of wind conditions. A strong circumpolar flow is a characteristic feature of the Southern Ocean with the southern tip of South America being the only barrier in the latitude band from

47° S to 63° S. Low-pressure systems embedded in this flow usually pass through the Drake passage south of Punta Arenas
causing prevailing wind directions between south-west and north-west. A comprehensive description of the meteorological
conditions is provided in Radenz et al. (2021).

Aeolus overpasses considered for the validation were on Wednesdays, ca. 23:26 UTC on the ascending orbit, and on Thursdays, at around 09:56 UTC on the descending orbit. For the presented study, the scanning Doppler cloud radar has been used
for the long-term validation and is thus explained in more detail in Section 2.2. The Doppler lidar performed scans for the
horizontal wind as well, but due to the very low amount of particles in Punta Arenas, the performance during the scans was not
optimum for the Aeolus validation. Mostly, wind retrievals were restricted to the local boundary layer. But due to the relatively
long distance to the Aeolus ground track (often more than 50 km) and the complex orography, the data were not useful for the
Aeolus validation.

### 2.1.2 Leipzig, Germany

At the ACTRIS (Aerosol, Clouds and Trace Gases Research Infrastructure) site of Leipzig, Germany (51.35° N, 12.43° E),
Aeolus Cal/Val activities were focused on dedicated radiosonde launches (see Sec. 2.2.2). These launches took place for the
ascending orbit of Aeolus on Friday evening at around 16:50 UTC.

Leipzig is located in central Europe being in the intermediate state between maritime and continental climate. Prevailing winds
are usually westerlies, but due to wave activities winds from all directions can be observed. Leipzig is located in the low-land
area. No orographical obstacles are around the city, making it a perfect location for the validation of Aeolus.

## 2.2 Instrumentation

### 2.2.1 Scanning Doppler cloud radar

Continuous measurements were conducted with a 35 GHz Doppler cloud radar of type Metek MIRA35 (Görsdorf et al., 2015).
Once per hour, the stare mode (vertical profiling) was interrupted for a Range-Height-Indicator (RHI) and Plan Position Indicator (PPI, also called VAD - Variable Azimuth Display) scan from minute 29 to 36 of each hour. The PPI scan started around
minute 35 and lasted for 60 seconds and covered one full 360° rotation made in 6° steps. Only the PPI scans are considered for
the horizontal wind retrieval, which were performed at an elevation angle of $\varepsilon = 85°$. A pulse repetition frequency of 5000 Hz
gives a maximum unambiguous radial velocity of $10.56\,\mathrm{ms^{-1}}$, while the range resolution of 31.17 m is determined by the pulse
length of 208 ns. Frequent cloud occurrence over Punta Arenas makes this instrument a perfect tool for retrievals of horizontal
wind profiles, particularly during austral winter (Seifert et al., 2020; Radenz et al., 2021). The methodology for retrieving wind
information from scanning Doppler remote-sensing instruments is described in more detail in Sec. 3.1.

### 2.2.2 Radiosonde

Radiosondes of type Vaisala RS41 (Jauhiainen et al., 2014; Jensen et al., 2016) were launched at Leipzig each Friday for
the regular Aeolus evening overpass (on its ascending orbit) since May 2019. The launch time of the radiosondes has been

at 16 UTC, thus ca. 50 minutes before the Aeolus overpass, to have a good coverage of the atmospheric conditions up to about 25 km. Usually, the complete ascent up to the burst height is about 2 hours and thus perfectly centered around the overpass time to have the best temporal coverage as possible. We therefore assume that the horizontal drift of the radiosonde does not introduce a systematic bias to our statistical validation analysis. The RS41 delivers profiles of temperature, humidity, pressure, wind speed and direction. The uncertainty for the wind products is estimated to be between 0.4 and 1 $\mathrm{ms^{-1}}$ for the wind velocity and 1° for the wind direction based on calculations of the Global Climate Observing System Reference Upper-Air Network (GRUAN, Dirksen et al., 2014). Even though these estimations are based on Vaisala radiosonde type RS92, there is no significant difference in the uncertainty between both radiosonde types as they are based on the same technique to derive wind velocity and direction (Jensen et al., 2016). A bigger gap in coverage occurred during winter 2020/2021 with only sporadic radiosondes (the reason was local access restriction due to COVID-19), but in total more than 125 launches could be completed. These radiosonde profiles were not assimilated so that they can serve as an independent reference for Aeolus products.

For Aeolus overpasses at Punta Arenas, dedicated radiosondes were launched irregularly. The radiosonde type deployed in Punta Arenas was Lockheed Martin LMS6 and delivered also profiles of temperature, humidity, pressure, wind speed and direction. In total, 41 radiosondes were launched during the 3 years campaign. Due to the irregularity of the launches, the radiosonde data was used only for case study analysis and thus also mainly for the validation of the cloud-radar-derived winds.

## 3 Methodology

The methodology how Aeolus retrieves the HLOS is described in many other publications, e.g. in detail in Krisch et al. (2022); Rennie et al. (2021) but also in Baars et al. (2020b); Weiler et al. (2021b); Chou et al. (2022); Witschas et al. (2022); Bley et al. (2022) and references therein. However, the methodology how to retrieve horizontal wind vector from ground-based Doppler instruments, like the Doppler cloud radar at Punta Arenas, is not straight forward as several methodologies exists. Here, we give a short overview, which methods are used and, thus, how the HLOS is retrieved from the cloud radar (Section 3.1) and about the general validation strategy for Aeolus (Section 3.2).

### 3.1 Retrieving horizontal wind profiles from radar

While the remote-sensing instruments of TROPOS in Punta Arenas are usually operating in stare mode (vertical profiling), regular PPI scans have been performed with the Doppler cloud radar to obtain the horizontal wind vector. For these scans, the radar is rotated around azimuth $\alpha$ with a fixed elevation angle $\varepsilon$ (which is set to $85\,°$). A sketch showing the different measurement modes is provided in Fig. 2, left.

The measured line-of-sight (LOS) Doppler velocity $v_{\mathrm{LOS}}$ at the range $R$ and azimuth angle $\alpha$ is retrieved as the mean of the measured Doppler velocities for a given range band $\Delta r$. The Root Mean Square (RMS) of this distribution is used to calculate the uncertainty of $v_{\mathrm{LOS}}$. In the following, we use the notation that is given in Päschke et al. (2015), which is shown in Fig. 2, right, and neglect that all variables are a function of the range $R$ to allow better reading. The final result is of course depending

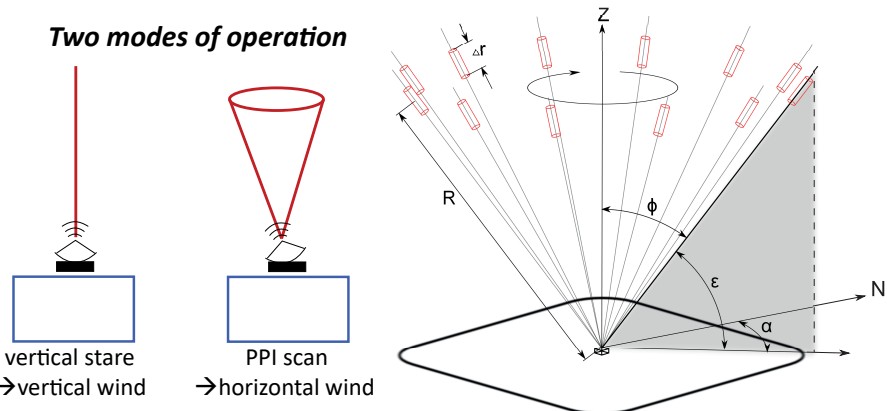

**Figure 2.** Left: Sketch of the different operating modes of the Doppler cloud radar. Stare mode is used for retrieval of vertical wind speed, while PPI/VAD scans are used to retrieve profiles of horizontal wind speed and direction. Right: Scanning geometry and nomenclature for the PPI/VAD scans as used in this work. The sketch was presented in Päschke et al. (2015) under Creative Commons Attribution 3.0 License and is shown here with permission of the authors.

on $R$ and thus gives a vertical profile.

In line with the standard approach of deriving horizontal wind from PPI scans (Browning and Wexler, 1968; Päschke et al., 2015), the mean horizontal wind speed $v_{\text{advection}}$ can be approximated by fitting the measured $v_{\text{LOS}}$ with a trigonometric function of the azimuthal coordinate of the scan corrected for positioning errors ($\alpha_{\text{corrected}}$):

$$v_{\text{projection}}(\alpha_{\text{corrected}}) = v_{\text{advection}} \cdot \cos(\alpha_{\text{corrected}} - \alpha_{\text{wind}}) + B + \sigma. \tag{1}$$

This formula gives the horizontal wind direction $\alpha_{\text{wind}}$ and the horizontal wind speed $v_{\text{advection}}$, with $v_{\text{projection}}$ being the horizontal component projected from $v_{\text{LOS}}$:

$$v_{\text{projection}} = \frac{v_{\text{LOS}}}{\cos(\varepsilon)}. \tag{2}$$

The term $\sigma$ in Eq. (1) reflects the remaining variation and is aimed to be minimized. For the fit procedure, $v_{\text{advection}}$, $\alpha_{\text{wind}}$, and $B$ are dependent variables, chosen to minimise $\sigma$ which is the remaining residual. The extra term $B$ stands for the contribution of two factors to the measured Doppler velocity: The divergence in the wind field and the vertical component of the average wind velocity. Both effects are neglected within the following analysis, as it is also done for the Aeolus HLOS retrieval.

Three different fit methodologies are used to derive the horizontal wind vector. The first one is equivalent to the methodology described in Päschke et al. (2015):

1. A least square regression is applied to fit $v_{\text{projection}}$ and $\alpha_{\text{corrected}}$ considering also their uncertainties. This method is subject to Doppler folding (Ray and Ziegler, 1977). This means, $v_{\text{LOS}}$ that exceeds the Nyquist velocity will appear as smaller measured velocity in the opposite direction. Usually, this effect will result in a poor fit quality with a high residual, as the measured $v_{\text{LOS}}(\alpha_{\text{corrected}})$ and thus $v_{\text{projection}}(\alpha_{\text{corrected}})$ will not approximate a trigonometric function, and can thus be discarded.

The other two fit methods that are applied to retrieve wind velocities from the raw Doppler velocity data are based on the method by Tabary et al. (2001), which uses the approximation of an azimuthal derivative of the velocity distribution. This method is performed in two different ways:

2. The horizontal wind vector profile is retrieved from the gradient $\partial v_{\text{LOS}}/\partial \alpha_{\text{corrected}}$ which is approximated by overlapping piece-wise linear fits centred around each point of the initial distribution as recommended by Tabary et al. (2001). This procedure is the second method and is usually consistent with the first method, but may lead to higher standard deviations due to removal of data points and the extra stages of calculation. Conversely, when Doppler folding occurs, this method is able to fit to a transformed version of the data with much higher accuracy.

3. The third method is applied because processing large numbers of linear fits as for the second method can sometimes be numerically unstable. This backup method applies the direct differences between consecutive values divided by the azimuthal distance. This approach is consistent with the former ones but leads to correspondingly higher errors because it excludes the averaging that occurs with the linear fit procedure. On the other hand, if the previous method fails to converge, this (third) Doppler-folding-safe methodology can be applied to derive the horizontal wind vector.

All three methods are performed for each range $R$ to calculate the horizontal wind vector. In a final step, a best estimate is computed, which selects the method with the lowest error out of the three methods. In the data set, the retrieval results from all three methods plus the best estimate is stored. This best estimate is then used for the comparison with the Aeolus winds, however not considering cloud-radar-derived HLOS winds with an error higher than $10\ \text{ms}^{-1}$.

## 3.2 Aeolus validation strategy

For the validation of Aeolus, we focus on the L2B wind products obtained by the Rayleigh methodology in clear air, called Rayleigh clear winds, and with the Mie methodology in clouds, called Mie cloudy winds. A more thorough description of the different products can be found, e.g., in the product description document (de Kloe et al., 2023) or in Baars et al. (2020b). If not otherwise stated, from now on we use the term "Rayleigh" for the Rayleigh clear wind products and the term "Mie" for the Mie cloudy wind products not considering the theoretically available Rayleigh cloudy and Mie clear winds. Rayleigh winds are delivered at 87 km horizontal resolution, while the Mie wind resolution has been mainly at 15 km (setting flexible, see Table 1).

We analyse all Aeolus-derived Horizontal Line-of-Sight (HLOS) wind speeds (i.e., at different altitudes) from the Rayleigh and Mie products according to their mean coordinates (in the center of the horizontal width) that are within a radius of 100 km[1] around the measurement site. The used radius is a good compromise between the number of available comparison data and the representativeness between the two different measurements as shown in other studies (Geiß et al., 2019; Cossu et al., 2022). Accordingly, two overpasses per week for each station fulfil these conditions and have been suitable for validation:

– Punta Arenas: Wednesdays at 23:26 UTC and Thursday at 09:56 UTC,

---

[1]120 km in Punta Arenas after orbit shift in June 2021.

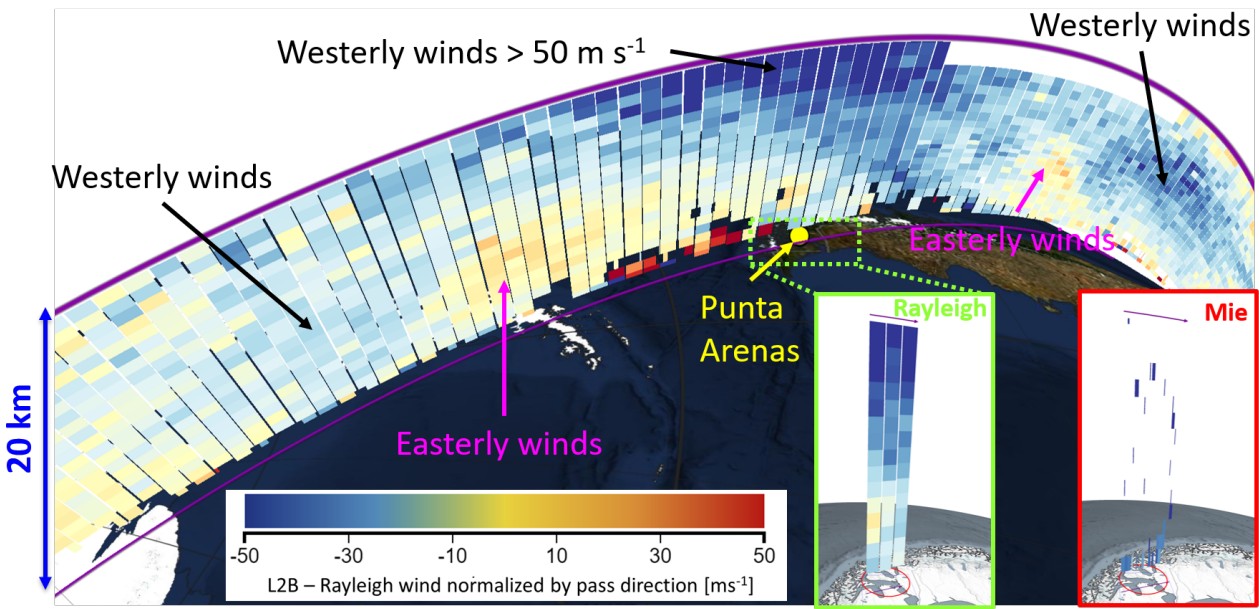

**Figure 3.** Example for L2B wind curtain around Punta Arenas, Chile, on 30 September 2020 visualized with VIRES (Santillan et al., 2019). The Rayleigh wind product is shown for the whole curtain. The Mie wind product (red box) and the Rayleigh wind products (green box) 100 km around Punta Arenas are shown in the lower right.

– Leipzig: Fridays at 16:50 UTC and Sundays at 05:29 UTC.

The performed orbit shift has not changed the considered overpasses for the stations, but the mean distance from the ground site to the Aeolus wind curtains has changed. Before the orbit shift, this mean distance was 17 km and 42 km on Fridays and Sundays at Leipzig, and 27 km and 75 km on Wednesday and Thursdays at Punta Arenas, respectively[2]. After the orbit shift, it changed to 33 km and 93 km at Leipzig, and 75 km and 85 km at Punta Arenas, respectively. As the orbit of Aeolus is slightly varying and the distances given here are only mean values, some of the overpasses at Punta Arenas were outside the 100 km criterion so that we adjusted the radius to 120 km. For Leipzig this was not needed, as only Friday overpasses were considered. For a better understanding of the general procedure, an example of the Aeolus Rayleigh wind profiles over Punta Arenas is shown in Fig. 3. On 30 September 2020, strong westerly winds (blueish colors) occurred over Punta Arenas at altitudes above 5 km. Closer to the South Pole, easterly winds (reddish colors) were prevailing. A patchy wind speed pattern was observed close to Punta Arenas near the ground, caused by cloud contamination of the Rayleigh winds. Given the example in Fig. 3, one sees that depending on the actual track of Aeolus, 1–3 wind profiles fulfil the criterion of being within 100 km radius of the observational site (see green box in Fig. 3). Considering 15 km horizontal resolution for the Mie product since 5 March 2019 (before the resolution was 87 km), one can have up to 13–20 "Mie winds" for one altitude range within 100 km of the ground-based location (see red box in Fig. 3). For the validation of Aeolus products, the temporally closest ground-based cloud radar

---

[2]calculated by ESA based on Orbit Scenario Files

measurement has been used allowing for a maximum-time-difference threshold of 1 hour. This threshold shall ensure similar atmospheric conditions for the validation. For the radiosonde data, such a temporal constraint is not needed as the radiosondes have been launched directly for the Aeolus overpass.

Furthermore, we converted the wind speed $v_{\mathrm{ref}}$ and direction $\varphi_{\mathrm{ref}}$ obtained with the reference instruments (subscript ref, i.e., cloud radar and radiosonde) to Aeolus-like HLOS winds $v_{\mathrm{ref_{HLOS}}}$ with the equation described in Baars et al. (2020b):

$$v_{\mathrm{ref_{HLOS}}} = v_{\mathrm{ref}} \cdot \cos(\varphi_{\mathrm{Aeolus}} - \varphi_{\mathrm{ref}}). \tag{3}$$

$\varphi_{\mathrm{Aeolus}}$ is the azimuth angle of Aeolus, which is obtained from the Level 2B data and differs depending on global position. The uncertainties of the ground-based observations were propagated forward. The derived ground-based profiles of HLOS
wind were then vertically averaged to the Aeolus-range bin thickness (500 to 2000 m, mostly higher resolution near ground and coarse resolution at high altitudes: Stoffelen et al., 2005; Straume et al., 2020; ESA, 2020; Bley et al., 2022) to allow a one-to-one comparison. This means, we do not aim to discuss here the small-scale wind variability within the relatively large Aeolus range bins but rather concentrate on performance of the space-borne instrument.

During the lifetime of Aeolus, several algorithm versions of the processing chain (so-called baselines) were released and
applied, some of them in operational mode, some of them to reprocess parts of historical Aeolus data. Thus, for certain dates in the Aeolus data set, several versions exist (processed with different baselines), while for other periods only one baseline was applied. An overview of the different baselines of Aeolus covering the observational period of our ground-based reference measurements (i.e., up to autumn 2022) is given in Table 1. Two major steps for boosting the performance of Aeolus were made. With Baseline 04, the so-called hot pixel correction (Weiler et al., 2021a) was introduced. Before that, single pixels
on the ACCD of Aeolus had a higher dark current and thus biased the retrieved winds. A second important step was the introduction of a correction with respect to changes in the telescope temperature of Aeolus (M1 temperature correction, Weiler et al., 2021b). This correction was implemented with Baseline 09 and should have brought a significant improvement of the performance of the Aeolus winds.

The switch from laser FM-A to laser FM-B was performed from 12 June 2019 until 28 June 2019 and led to Baseline 05.
However, a new response calibration needed to be applied, which was obtained in August 2019 and led to Baseline 06. The FM-B data before that date was then reprocessed. In June 2021, the orbit of Aeolus was shifted to favor the ground-based observations in Cabo Verde during the Joint Aeolus Tropical Atlantic Campaign (JATAC, Fehr et al., 2021). Therefore, mean horizontal distances of the Aeolus wind products to the ground-based reference stations changed and as a result we increased the maximum radius for Punta Arenas to 120 km to still be able to obtain two overpasses per week (as already discussed
above).

Accounting for changes in units for the uncertainties within the Aeolus products between different baselines, all values for Aeolus horizontal wind speed and errors were transformed into $\mathrm{ms^{-1}}$. Beside this unit correction, all baselines were treated equally. Furthermore, next to the provided validity flag within the Aeolus wind products, additional quality measures, i.e., error thresholds for Mie and Rayleigh winds (5 $\mathrm{ms^{-1}}$ and 8 $\mathrm{ms^{-1}}$, respectively), have been applied. This means that wind products
flagged valid but with an error higher than these thresholds were discarded. This approach is consistent with DISC/ESA

**Table 1.** Overview of the different algorithm versions (called baseline) for the processing of the Aeolus data together with some important additional information.

| Baseline | Period | Start date for operational processing | Additional info |
|---|---|---|---|
| B02 | Sep 2018 – May 2019 | 8 Sep 2018 | Mie wind horizontal resolution from 87 km to 15 km in March 2019 |
| B03 | May 2019 – June 2019 | 16 May 2019 | |
| B04 | June 2019 | 14 June 2019 | Hot Pixel correction (Weiler et al., 2021a) |
| B05 | June 2019 – September 2019 | 28 June 2019 | Switch to laser FM-B |
| B06 | June 2019 – October 2019 | 5 September 2019 | Adapted response calibration for FM-B |
| B07 | October 2019 – April 2020 | 31 Oct 2019 | |
| B08 | April 2020 | 2 April 2020 | Unit change of HLOS error from $\mathrm{ms}^{-1}$ to $\mathrm{cms}^{-1}$ |
| B09 | April 2020 – July 2020 | 20 April 2020 | M1 temp. correction (Weiler et al., 2021b), public data release |
| B10 | June 2019 – Oct 2020 | 2019 reprocessed since 9 July 2020 operational | reprocessed FM-B data set included in Baseline 10 covering June to December 2019 |
| B11 | June 2019 – May 2021 | 2019 reprocessed since 8 October 2020 operational | reprocessed FM-B data set included in B11 covering the period June 2019 to 10 October 2020 |
| B12 | May 2021 – Dec 2021 | 26 May 2021 | Orbit shift in June 2021 |
| B13 | Dec 2021 – Mar 2022 | 6 December 2021 | |
| B14 | Mar 2022 – Sep 2022 | 29 March 2022 | |

recommendations (e.g., Witschas et al., 2020; Rennie and Isaksen, 2020) and studies by other Cal/Val teams (e.g., Guo et al., 2021; Chen et al., 2021; Iwai et al., 2021; Abril-Gago et al., 2022b). However, it needs to be mentioned, that for future validation studies of re-analysed wind products, other quality control approaches should be considered as the quality of the estimated error will also change with changing baseline and thus also its applicability as an additional quality control parameter as, e.g., discussed in Lux et al. (2022b).

For the statistical analysis presented in Section 5 and 6, the Rayleigh and Mie wind products were treated separately. To obtain statistical metrics, a straight-line fit between the ground-reference winds and the Aeolus winds using a orthogonal distance regression (ODR) to include the effects of errors has been computed. We also created histograms of the deviations (reference wind minus Aeolus wind) in the range of -15 to +15 $\mathrm{ms}^{-1}$ (with higher velocities being assigned to the outside bins) to check for a Gaussian distribution shape. A walk-through example for the statistical analysis is given in Section 5, after the general validation strategy based on case studies is discussed next.

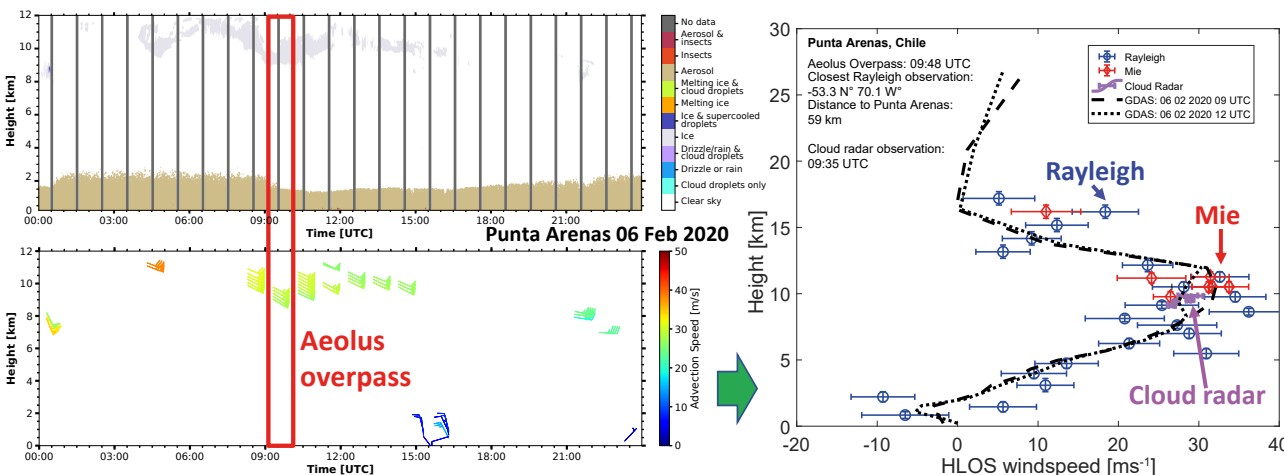

**Figure 4.** Schematic overview on the methodology used in this study for the example of 6 February 2020 in Punta Arenas. Top, left: Cloudnet Target Categorization obtained from the combination of vertically measuring (in stare mode) cloud radar, ceilometer and microwave radiometer. Bottom, left: Wind speed and direction indicated by arrows as retrieved with the Doppler cloud radar scans at minute 35 of each full hour in regions of cloud occurrence. For a better visibility, only every second wind barb is shown for measurements above 8 km. Right: Comparison of the HLOS winds from Doppler cloud radar to Aeolus products (Mie and Rayleigh, Baseline 11) for the closest overpass. GDAS model winds are shown for comparison as well.

## 4 Case studies

To illustrate the validation strategy and discuss the potentials and drawbacks, we present two interesting case studies performed for Punta Arenas in the following.

### 4.1 Punta Arenas - 6 February 2020

A schematic overview on how winds are retrieved from the ground-based observations and then compared to Aeolus wind products is shown in Fig. 4 for the case of 6 February 2020, representative for the Southern Hemispheric summer. The atmospheric conditions above Punta Arenas on this day are presented in Fig. 4, top, left, by means of the Cloudnet Target Categorization (Illingworth et al., 2007; Tukiainen et al., 2020; Radenz et al., 2022) derived from the vertically starring active remote sensing instrumentation (cloud radar and ceilometer) and the passive microwave radiometer. During this day, a nearly cloud-free aerosol layer from ground up to 1.5 km altitude was observed with enough particles to be identified by Cloudnet Target Categorization (dark yellow). Partly, the cloud radar observed a return signal within this aerosol layer, which is attributed to insects (not shown). Between 2 and 8 km, clear-sky conditions (white color) were found, while ice clouds (dark grey) occurred sporadically above 8 km altitude. Horizontal wind vector observations retrieved with the Doppler cloud radar from the hourly scans at minute 35 of each hour were therefore available in atmospheric regions where clouds were existing, see Fig. 4, lower, left.

The Aeolus overpass on this day was at 09:48 UTC indicated by the red rectangle in Fig. 4, left. Thus, the temporally closest wind profile from the Doppler Cloud radar (at 09:35 UTC) plus the HLOS profiles extracted from GDAS1[3] data for 9 UTC and 12 UTC were used to compare with the Aeolus products (Mie and Rayleigh) as can be seen in Fig. 4, right. The closest distance between the Aeolus ground track and the ground site was 59 km on this day.

In this example comparison, the advantages and drawbacks of the used reference instrument becomes clear. The cloud radar is only able to retrieve winds in regions where clouds are existent (on this day between 8 and 12 km) and no information can be obtained in clear-sky regions. In regions of clouds, however, the winds can be obtained with high vertical resolution and high quality. The GDAS-derived vertical profile of HLOS is available for all atmospheric states (independent of cloud occurrence, clear sky etc.) but in coarser resolution. As GDAS data are a result of data assimilation, it is no direct validation measure and therefore shown only for consistency checks and not for Aeolus validation itself.

According to Fig. 4, right, the cloud-radar-derived HLOS wind taken at 09:35 UTC provides values of 27 to 31 $\mathrm{ms^{-1}}$ in the cloudy region around 10 km. GDAS HLOS wind speeds in this altitude region change from about 33 $\mathrm{ms^{-1}}$ to 28 $\mathrm{ms^{-1}}$ from 9 to 12 UTC, respectively. Due to the good agreement of the both data sources with respect to the HLOS, the validity of the cloud-radar derived winds can be assumed. Aeolus-derived wind profiles within a radius of 100 km were available in clear air (Rayleigh winds) and at top of clouds (Mie winds). The Mie winds available at 10 to 12 km indicate, thus, the presence of clouds at similar altitudes at the Aeolus track (more than 59 km away from the ground site). If the cloud deck would be persistent and optically thick over the whole horizontal Aeolus track, no Aeolus winds would be available below the altitude of the clouds due to the strong light attenuation within the cloud. As this is not the case because Rayleigh winds are available down to the surface, a broken cloud deck and/or optically thin clouds between 10 and 12 km with clear sky below and above can be considered in the Aeolus observation - which is in excellent agreement with the atmospheric scene observed over Punta Arenas by LACROS (Fig. 4, top, left).

On this specific day in austral summer 2020, a good agreement between the Mie winds and the cloud-radar-derived winds were obtained at an altitude of around 10 km. Also, the delivered Rayleigh winds in this altitude region agree well with the radar and also with GDAS. The coexistence of Rayleigh and Mie winds in one altitude range is possible because of the broken cloud deck Rayleigh and Mie winds can coexist (Reitebuch et al., 2018; Rennie et al., 2020; de Kloe et al., 2023) and all single Aeolus products within the defined horizontal radius of 100 km are considered. In contrast, at 17 km height, GDAS and Aeolus disagree for the only one Mie observation there. The reason for that is yet unclear but might be related to atmospheric heterogeneity, uncertainties of GDAS (which are not provided) or a misclassification of Aeolus as 17 km is usually well above the local tropopause and thus no clouds are expected (and not seen in the ground observations). However, smoke occurrence from Australian bush fires at this altitude range cannot be ruled out (Ohneiser et al., 2020), but the ground based PollyXT lidar observations do not show a significant enhanced backscatter at this region later the day, when the clouds at around 10 km disappeared, which would explain a misclassification by Aeolus. Thus, the obviously misclassified Mie wind observation should be revisited in the context of the validation of future re-processed data. Rayleigh winds show also deviations around 17 km towards higher HLOS wind speed in accordance to the Mie observations.

---

[3]Global Data Assimilation System (GDAS), ARL Archive: GDAS1 data set, available at: https://www.ready.noaa.gov/gdas1.php

Below the cloud deck at 10 km, Rayleigh winds are partly matching the model data (GDAS), but with a tendency of higher Aeolus wind speeds down to around 5 km altitude. Sporadically, lower Rayleigh HLOS wind speeds were also observed. Deviations within the lowest 3 km might be caused by horizontal inhomogeneity within the 100 km radius around the ground-based station. For the statistics presented below in Section 5 ff., we use wind derived with the Doppler cloud radar and compare them to the equivalent Aeolus HLOS winds. For the example case presented here, this means that a comparison to Mie winds is possible for the height range around about 11 km, as this is the only region for which cloud-radar-derived winds and Aeolus Mie winds coexist. Rayleigh wind comparisons can be done at the same height range (reference measurements from the cloud radar). The regions between the ground and 9 km altitude and above 11 km cannot be covered for the comparison due to the missing ground-based measurement data. Radiosondes could cover this gap, but they were launched at Punta Arenas only irregularly for the Aeolus validation so that a meaningful long-term validation is not possible. We did not aim at a validation with model data, as this is done regularly at ECMWF (Rennie and Isaksen, 2020; Rennie et al., 2021) and by other validation activities (e.g., Martin et al., 2021; Liu et al., 2022; Chen et al., 2021; Hagelin et al., 2021; Rani et al., 2022). Instead, we concentrate on the direct measurements made from ground.

## 4.2 Punta Arenas - 18 August 2021

The second case study from Punta Arenas presents an observation from the Southern Hemispheric winter. At this day, besides the Doppler radar, also a local radiosonde launch was available.

The atmospheric conditions on this day were remarkable as presented in Figure 5. While typical HLOS wind speeds between 5 and 20 $\mathrm{ms}^{-1}$ were observed in the troposphere, a steady increase in wind speed was observed above the tropopause (at ca. 9.5 $\mathrm{km}$) leading to a maximum wind speed measured by the radiosonde of more than 100 $\mathrm{ms}^{-1}$. The Aeolus-derived wind speed profile (overpass at 23:27 UTC, with the closest distance of 47 $\mathrm{km}$ to Punta Arenas) is in agreement with the GDAS (21 and 00 UTC) and the radiosonde data (launched 23 UTC), and thus gives confidence of its reality. The horizontal extent of the strong winds around Punta Arenas can be seen in Fig. 5, bottom, right, which shows the Aeolus Rayleigh HLOS wind speed. It demonstrates the potential of Aeolus to detect such features.

The comparison of Aeolus derived HLOS winds with the radiosonde, cloud radar (23:35 UTC) and GDAS is shown in Fig. 5 and reveals that the Rayleigh product above 10 km follows the observed radiosonde wind profile and GDAS products. A considerable deviation was observed only at the top most Aeolus range bin (around 24 km). There, the wind speed measured by Aeolus was considerably lower compared to the radiosonde wind. The reason is not yet clear, but might be simply due to the strong drift of the radiosonde because of the high wind speed. Below the tropopause at ca. 9.5 $\mathrm{km}$, the Aeolus Mie winds and Rayleigh winds agreed mostly with the radiosonde. The GDAS data, however, show a significant variation at 8 km between the two profiles at 21 UTC and 00 UTC. This behaviour implies that fast changes in HLOS, i.e., in wind speed and direction, have taken place in this atmospheric region. It furthermore gives confidence, that the 1-hour time window for validation is a reasonable time period for the wind validation.

Usually, two Aeolus Rayleigh wind observations (of 87 km horizontal length in case of Rayleigh) per Aeolus height bin exist in the validation radius of 100 km around the ground station when using the centre coordinates of the wind products and are

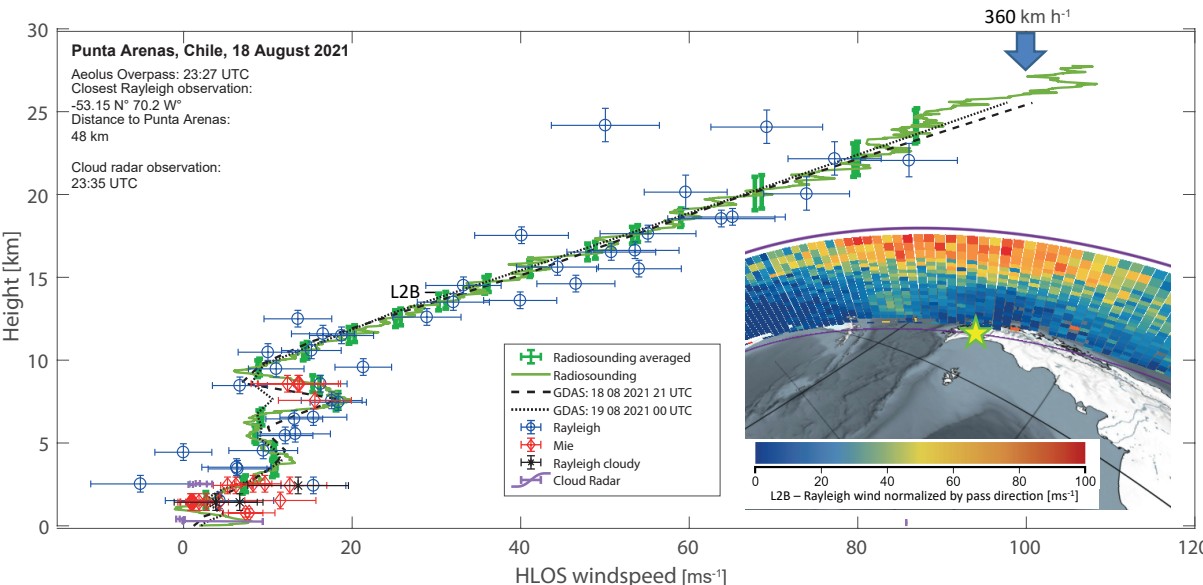

**Figure 5.** Comparison of radiosonde, Doppler cloud radar, and GDAS HLOS winds to Aeolus products on 18 August 2021 in southern hemispheric winter conditions. Radiosonde vertical error bars indicate the mean wind speed of the radiosonde averaged to the Aeolus range bins, Aeolus vertical error bars indicate the respective range-bin extent. The Aeolus Rayleigh wind curtain for the analyzed overpass as visualized with VIRES (Santillan et al., 2019) is shown in the lower right.

thus considered for the validation. Having a look at the HLOS observations at around 4.7 km, one sees that one wind product of Aeolus fits very well to the reference wind profiles, while the other one shows considerable deviations (around 10 $\mathrm{ms}^{-1}$ lower HLOS). This finding implies regional variations in the wind pattern. As a consequence, the observed outliers in the Aeolus Rayleigh winds at 3 and 4.7 km can be attributed to horizontal (and thus also temporal) heterogeneity in the wind field. This behaviour shows the difficulty in comparing Aeolus HLOS winds to the ground-based observations, because a perfect co-location in space and time can usually never be achieved. However, we consider that these meteorological variations do not lead to additional biases in the statistics presented in Section 5 ff., but are properly covered by the statistical methodologies in terms of random error.

## 5 Example for statistical validation: Baseline 11 at Leipzig and Punta Arenas

To obtain statistical measures for the performance of Aeolus and its algorithms, we analyzed the Aeolus HLOS data by Doppler cloud radar and radiosonde (from now on called reference instruments) as described above for the locations of Leipzig, Germany, and Punta Arenas, Chile. To illustrate that approach, the validation of Aeolus Baseline 11 products around Punta Arenas are shown in Fig. 6. The direct comparison (left column) and the frequency distribution of the deviation from the Doppler cloud radar values (reference minus Aeolus, right column) are shown for the Mie (top row) and Rayleigh winds (bottom row).

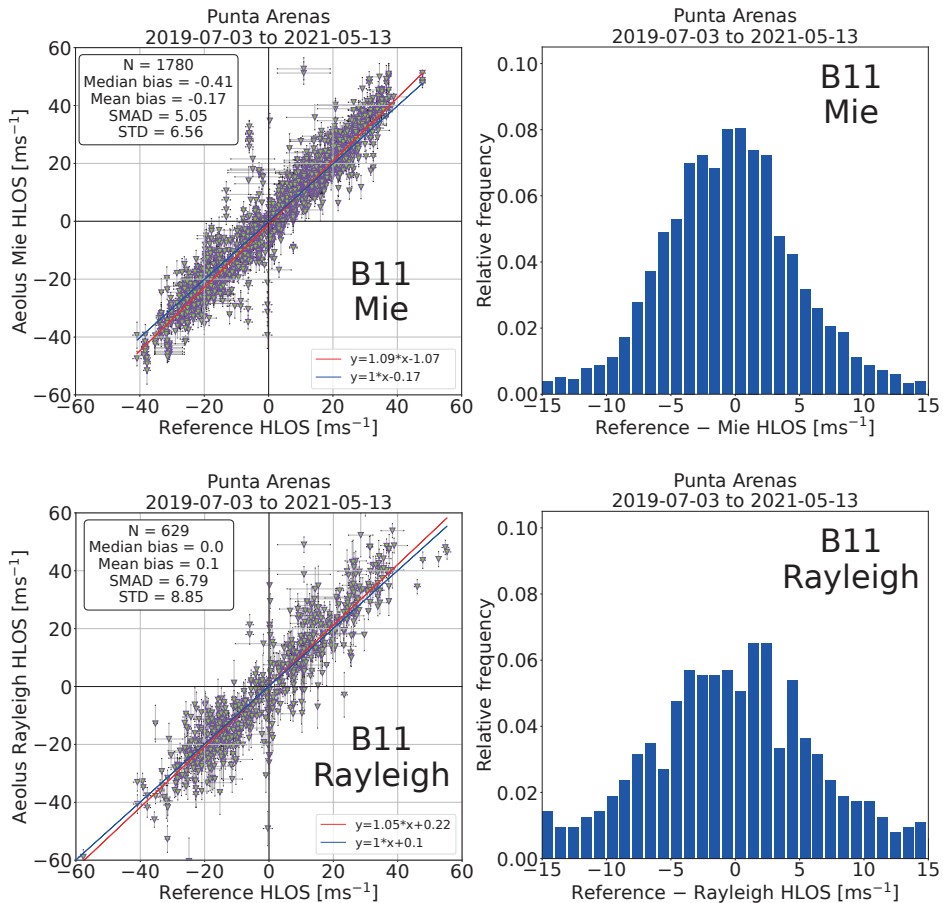

**Figure 6.** Long-term wind statistic for the Baseline-11-wind-validation based on ground-based Doppler cloud radar observations at Punta Arenas (ascending and descending orbit). Top: Aeolus Mie wind, bottom: Rayleigh wind, left: 1:1 statistic with respective measures (median and mean bias, scaled median absolute deviation (SMAD), and standard deviation (STD) provided in $\mathrm{ms^{-1}}$, and N indicating the number of samples), right: frequency distribution of differences between the two data sources.

A generally good agreement can be seen between Aeolus and the cloud radar being most of the time close to the one-to-one line and thus justifying the use of the orthogonal distance regression (ODR) for fitting the data.

More data points could be evaluated for the Mie winds (in total 1780) than for the Rayleigh winds (629), which is a logical consequence of the higher resolution of the Mie winds and the fact that the cloud radar derives winds only in regions with cloud occurrence. Thus, the validation is more meaningful with respect to Mie winds.

For the Mie wind, we obtained a slope of 1.1 with the ODR. When forcing the slope to be unity, the resulting offset is equal to the mean bias as expected for a Gaussian distribution. A median bias of $-0.41\ \mathrm{ms^{-1}}$ and a mean bias of $-0.17\ \mathrm{ms^{-1}}$ was derived (i.e., Aeolus measures less than the ground-based reference) together with a standard deviation of $6.6\ \mathrm{ms^{-1}}$ and

**Table 2.** Overview of the metrics obtained for the validation of Baseline 11 at Punta Arenas (left, all orbit types) and Leipzig (right, ascending orbit).

(a) Aeolus vs. cloud radar at Punta Arenas

| | Mie | Rayleigh |
|---|---|---|
| Number of points | 1780 | 629 |
| Slope | 1.09±0.01 | 1.05±0.01 |
| Median bias ($\mathrm{ms^{-1}}$) | −0.41 | 0 |
| Mean bias ($\mathrm{ms^{-1}}$) | −0.17 | 0.1 |
| Scaled MAD ($\mathrm{ms^{-1}}$) | 5.05 | 6.79 |
| Standard deviation ($\mathrm{ms^{-1}}$) | 6.56 | 8.85 |

(b) Aeolus vs. radiosonde at Leipzig

| | Mie | Rayleigh |
|---|---|---|
| Number of points | 1751 | 2361 |
| Slope | 0.93±0.01 | 0.97±0.01 |
| Median bias ($\mathrm{ms^{-1}}$) | −0.35 | −0.46 |
| Mean bias ($\mathrm{ms^{-1}}$) | 0.03 | −0.22 |
| Scaled MAD ($\mathrm{ms^{-1}}$) | 4.59 | 5.77 |
| Standard deviation ($\mathrm{ms^{-1}}$) | 5.34 | 6.75 |

a scaled median absolute deviation (MAD) of 5.1 $\mathrm{ms^{-1}}$. For the Rayleigh wind validation, we obtained respective values of 1.05 (slope), 0.1 $\mathrm{ms^{-1}}$ (mean bias), 0 $\mathrm{ms^{-1}}$ (median bias), 8.9 $\mathrm{ms^{-1}}$ (standard deviation) and 6.8 $\mathrm{ms^{-1}}$ (scaled MAD), see statistics box in the left column of Fig. 6, bottom.

In the following, we use the scaled MAD as an indicator for the random error, in analogy to the median bias for the systematic error often just called bias. The median values are less sensitive to outliers than the mean values, but are a valid measure for the uncertainties as long as the frequency distribution is of Gaussian shape. The philosophy of the use of the scaled MAD for Aeolus comparisons is in detail explained in Martin et al. (2021), Lux et al. (2022a), and Weiler et al. (2021a).

We also performed a $Z$-score analysis as described in Lux et al. (2022b) and found that, for example, with a $Z$ value of 3
for Baseline 10, 1.5 % of the Aeolus values are identified as outliers. However, in this publication we do not want to discuss the outliers of Aeolus wind products, but rather the performance of the publicly available wind data as a whole. Thus, we do not exclude outliers based on the $Z$-score analysis, but make a validation of the complete Aeolus data set by applying the recommended error thresholds of 8 $\mathrm{ms^{-1}}$ and 5 $\mathrm{ms^{-1}}$ for Rayleigh and Mie winds, respectively.

The frequency distribution of the differences (Fig. 6, right) shows a near-Gaussian form giving confidence that the statistical
measures described above can be applied. The obtained systematic errors of −0.41 $\mathrm{ms^{-1}}$ and 0 $\mathrm{ms^{-1}}$ and the random errors of 5 $\mathrm{ms^{-1}}$ and 7 $\mathrm{ms^{-1}}$ for Mie and Rayleigh products, respectively, for Baseline 11 validation at Punta Arenas are in line with results from other validation activities for this Baseline (Zuo et al., 2022; Geiß et al., 2022). An overview of the main key numbers from this statistic is given in Table 2. We performed the same statistical analysis with the radiosonde data from Leipzig for Baseline 11. The results are presented in Fig. 7. Given the fact that the radiosonde delivers wind data in both, clear and
cloudy skies, it becomes clear that this reference instrument is well suited for the Aeolus Rayleigh and Mie wind validation. As radiosondes are not limited to certain atmospheric targets, a coverage up to 25 km height could usually be achieved, allowing to validate all HLOS winds during an Aeolus overpass. Therefore, results are not confined to the cloud-laden troposphere like in Punta Arenas. Thus, the statistical analysis is more representative in terms of data points for the Rayleigh and Mie wind as can be seen in Fig. 7, left. In total, more than 1500 and 2000 data points could be used for validating the Aeolus Mie and

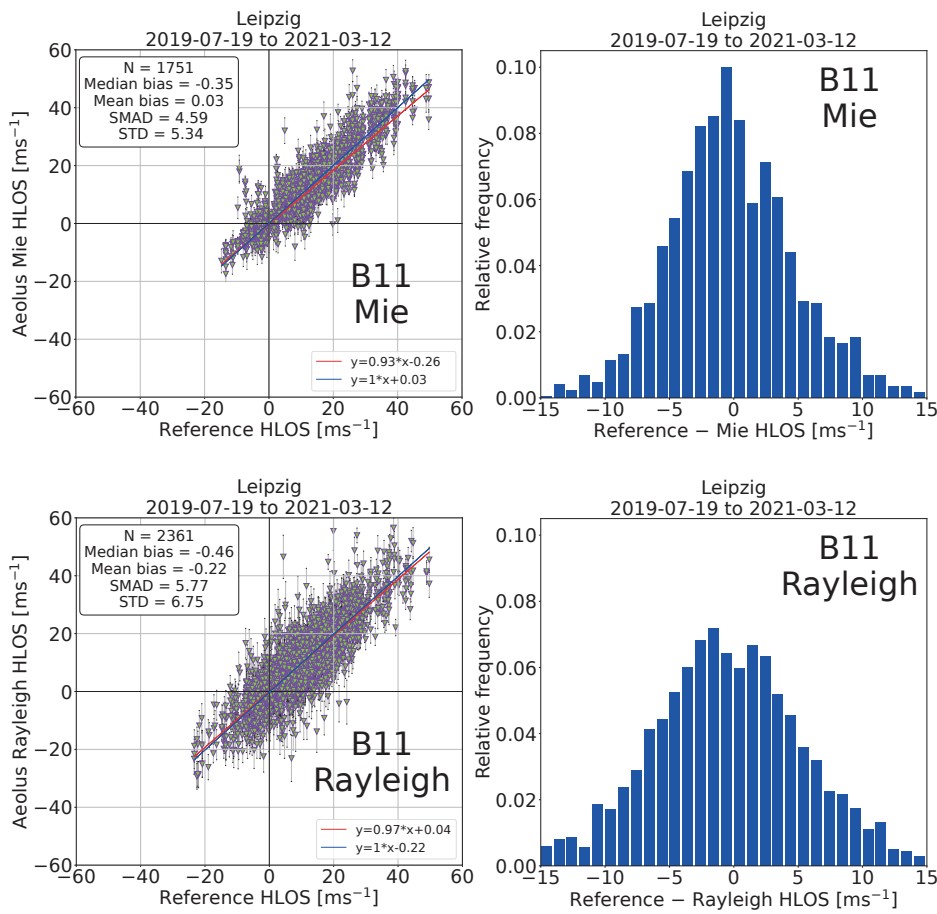

**Figure 7.** Long-term wind statistic for the Baseline 11 wind validation based on radiosonde launches at Leipzig. Top: Aeolus Mie wind, bottom: Rayleigh wind, left: 1:1 statistic with respective measures (median and mean bias, scaled median absolute deviation (SMAD), and standard deviation (STD) provided in $ms^{-1}$, and N indicating the number of samples), right: frequency distribution of differences between the two data sources. Observations on the ascending orbit node of Aeolus were considered only.

Rayleigh winds, respectively. The frequency distributions of the difference between the reference and Aeolus HLOS winds are of Gaussian shape for both Mie and Rayleigh winds and thus give again evidence for the validity of the applied statistical validation approach. The direct comparison (Fig. 7, left column) shows a generally good agreement with only sporadic outliers (e.g., $\approx 50\ ms^{-1}$ in the Aeolus Rayleigh wind product while the radiosonde delivered less than $10\ ms^{-1}$). The majority of data points are, however, near the 1:1 line and thus in good agreement. For the Mie winds, we obtained similar values as for Punta

Arenas in the Southern Hemisphere, with a systematic error of $-0.4\ ms^{-1}$ and a random error of $4.6\ ms^{-1}$. For the Rayleigh winds, we obtained a median bias of $-0.5\ ms^{-1}$ and a random error similar to the one for Punta Arenas with $5.8\ ms^{-1}$. Given the fact that more data points are available, the retrieved Rayleigh systematic error for Leipzig is more meaningful, even

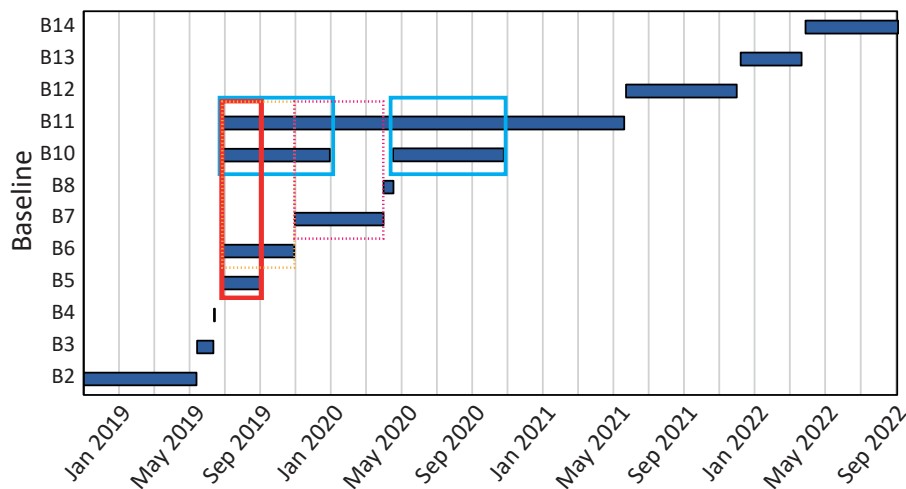

**Figure 8.** Overview of the different baselines (algorithm versions) that were used to process Aeolus data. The rectangles indicate the periods which have been used for the baseline comparison. The red rectangle refers to Section 6.1.1, the dark yellow dashed one to Section 6.1.2, the magenta dashed one to Section 6.1.3, and the light blue ones to Section 6.1.4.

though one has to consider that latitudinal and longitudinal dependencies of the systematic error have been discovered (Martin et al., 2021; Weiler et al., 2021b) and thus single locations like Leipzig and Punta Arenas are not completely representative for
the overall global performance of Aeolus. The so-called orbital bias was partly resolved with the M1 temperature correction (Weiler et al., 2021b). However, a leftover effect in this orbital bias oscillation cannot be ruled out. Especially, if one considers that for Leipzig we could only evaluate the ascending orbit, while for Punta Arenas we evaluated both orbit types.
We also performed a radiosonde-based validation for Punta Arenas, but too less data points from the very few radiosonde launches matching the evaluation criteria have been available so that the results are not statistically significant. We thus do not
further discuss this specific validation methodology for Punta Arenas with respect to the long-term analysis. A final overview of the obtained metrics for the validation of Baseline 11 is shown in Table 2. The same methodology has been applied to the other baselines and will be discussed below.

## 6   Aeolus validation

We performed the validation analysis for Punta Arenas (Doppler cloud radar) and Leipzig (radiosonde) for all available base-
430 lines, and thus for the time periods listed in Table 1. In Section 6.1, we analyze different product versions covering the same time period to assess the changes in product quality with changing baseline. Periods marked with red, light blue, magenta and orange rectangles in Fig. 8 are well appropriate for a baseline intercomparison. For the red-marked period from June 2019 – September 2019, products from four different algorithm versions are available covering already the FM-B era. The

dark-yellow-dashed-marked time period comprises Baseline 06, 10, and 11 from July 2019 to October 2019 and the magenta-dashed-marked period represents the comparison period for B07 to B11, which covers the time from November 2019 to April 2020. The light-blue-marked period reaches from June until December 2019 and from May 2020 to October 2020 and covers two different algorithm versions. In Section 6.2 we use the latest algorithm version (baseline) available for the analyzed time period to discuss the performance of the instrument during its lifetime.

## 6.1 Comparison of baselines

Due to the reprocessing efforts of the Aeolus team, there are certain periods in which Aeolus data are available for different baselines as shown in Fig. 8. This allows the validation of the improvements between the different baselines using the same reference data. However, a quantitative measure is not straightforward as due to quality control (QC) procedures etc., not the same amount of Aeolus wind data are available. Nevertheless, such a comparison gives a first inside into the improvement made by introducing new baselines.

### 6.1.1 B05, B06, B10, B11 comparison

To start with the analysis of the different baselines, we focus on the period from July to September 2019 for which data from four different baselines are available: B05, B06, B10, and B11 - see red rectangle in Fig. 8. The switch to laser FM-B had been already performed at this time. We analyzed this period with the reference data from Punta Arenas and Leipzig. The results for Punta Arenas are shown in Fig. 9 for the Mie (top) and Rayleigh (bottom) wind products.

As this period lasted two months, 16 overpasses in the Southern Hemispheric winter could be covered. The greatest differences can be seen for the Rayleigh winds. For Baseline 05, many outliers (data not close to the 1:1 line) have been observed, mainly at negative HLOS speeds, which led to a bias of $-8 \ \mathrm{ms}^{-1}$ for B05.The random error is as high as $20 \ \mathrm{ms}^{-1}$ for the Rayleigh winds for this baseline (B05). The poorly estimated error product of the Aeolus Rayleigh winds at this baseline might have caused that invalid winds observations have been flagged as valid. For the Mie winds, a systematic and random error of $-3 \ \mathrm{ms}^{-1}$ and $7 \ \mathrm{ms}^{-1}$ was found, respectively. With Baseline 06, the overall performance was improved on the cost of less valid data. This indicates already introduced improvement by the new response calibration which was needed since the switch to laser FM-B, but could be just obtained in August 2019 and thus two months after the switch. The fact that less valid Aeolus data was available might be caused by improved or more strict quality flags and error calculations (more wind observations flagged as invalid). Finally, less outliers are found for Rayleigh winds as seen in Fig. 9 (B06, Rayleigh). At this baseline (B06), the systematic and random error was as low as 0.4 (1.9) $\mathrm{ms}^{-1}$ and 5.1 (7.7) $\mathrm{ms}^{-1}$ for Mie (Rayleigh) winds, respectively. Of course, these numbers have to be assessed with care due to the low number of compared overpasses. Nevertheless, when looking at the Rayleigh wind metrics of Baseline 10, a significant improvement is found while also a slightly higher number of data are available. The introduction of the M1 temperature correction with Baseline 09 seems to have significantly improved the Rayleigh winds. Biases of 0.9 and 1.4 $\mathrm{ms}^{-1}$ have been detected with random errors of about 6 $\mathrm{ms}^{-1}$ for both B10 and B11, respectively. Also here, it holds that these numbers have to be taken with care due to the relatively low amount of data, but it definitely shows that algorithms have significantly improved and systematic and random errors are at a good level to allow the

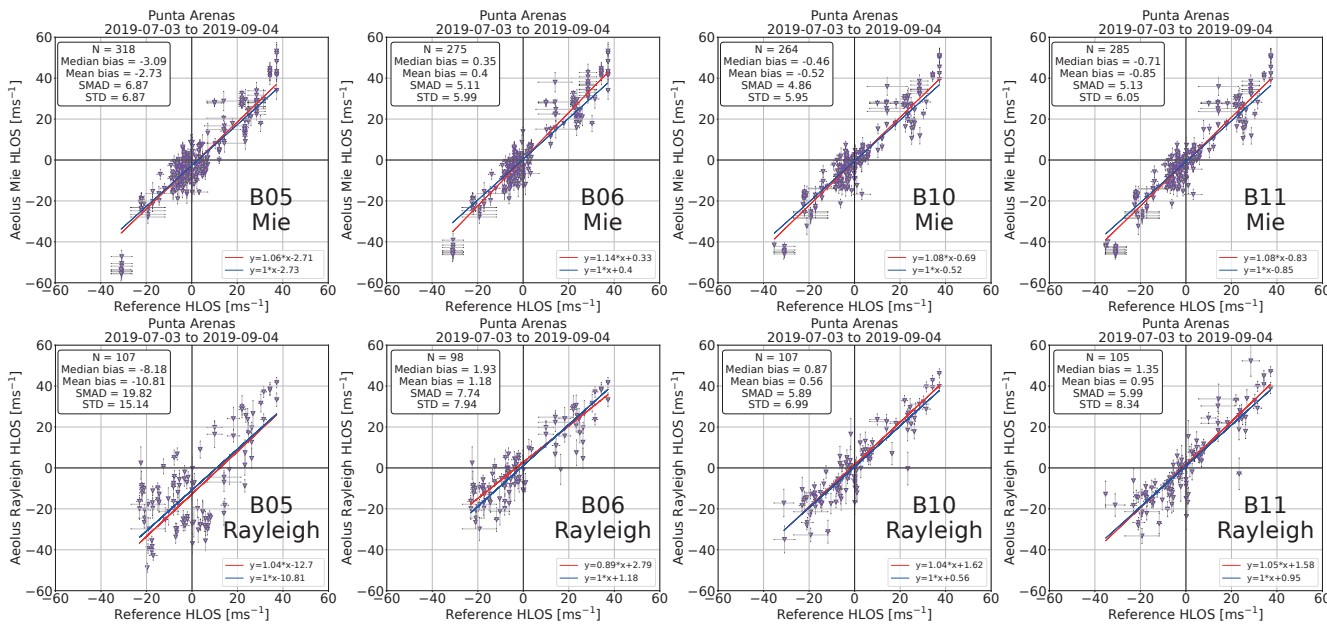

**Figure 9.** Aeolus performance as obtained by comparison to ground-based Doppler cloud radar at Punta Arenas for Baselines 05, 06, 10, and 11 in the period from July to September 2019 (ascending and descending orbit). Plots in analogy to Figures 6 and 7, left.

use of the Rayleigh wind products.

For the Mie winds, the improvement in performance is less evident compared to the Rayleigh products, caused by the fact that Mie winds were already much more reliable for B06 (bias of 0.4 $\mathrm{ms}^{-1}$ and random error of 5.1 $\mathrm{ms}^{-1}$). The number of available measurements also stayed nearly constant. The reason for having already reliable Mie winds for B06 is that according to Weiler et al. (2021b), the Mie winds are 10 times less affected by the M1 temperature variations than the Rayleigh products due to the technical nature of the different detection schemes (e.g., also Reitebuch, 2012).

With the introduction of B10 and B11, the median bias stays below absolute values of 1 $\mathrm{ms}^{-1}$ with random errors of about 5 $\mathrm{ms}^{-1}$ for the Mie products. For both wind types, the difference between B10 and B11 itself is less significant, most probably caused by the low amount of data that could be used for the comparison. An intense discussion on the B10 to B11 comparison is done later in Sec. 6.1.4 for a longer time period.

For Leipzig, a similar but not equal behaviour was observed as shown in Fig. 10. The Mie systematic errors were for all compared baselines with absolute values below 1.2 $\mathrm{ms}^{-1}$. The magnitude of the random error for the Mie winds was improved, but not as significantly as over Punta Arenas. In contrast, for the Rayleigh winds the bias was high at Baseline 05 (> 8 $\mathrm{ms}^{-1}$ in absolute values) and could be significantly reduced until B11 (< 0.2 $\mathrm{ms}^{-1}$). The major step forward concerning the random error for the Rayleigh winds was found since B06 due to the new response calibration function, leading to a decrease from 15 $\mathrm{ms}^{-1}$ to 4 $\mathrm{ms}^{-1}$. Interestingly, for the B05 Rayleigh winds (Fig. 10, lower, left), a divided distribution was found with much

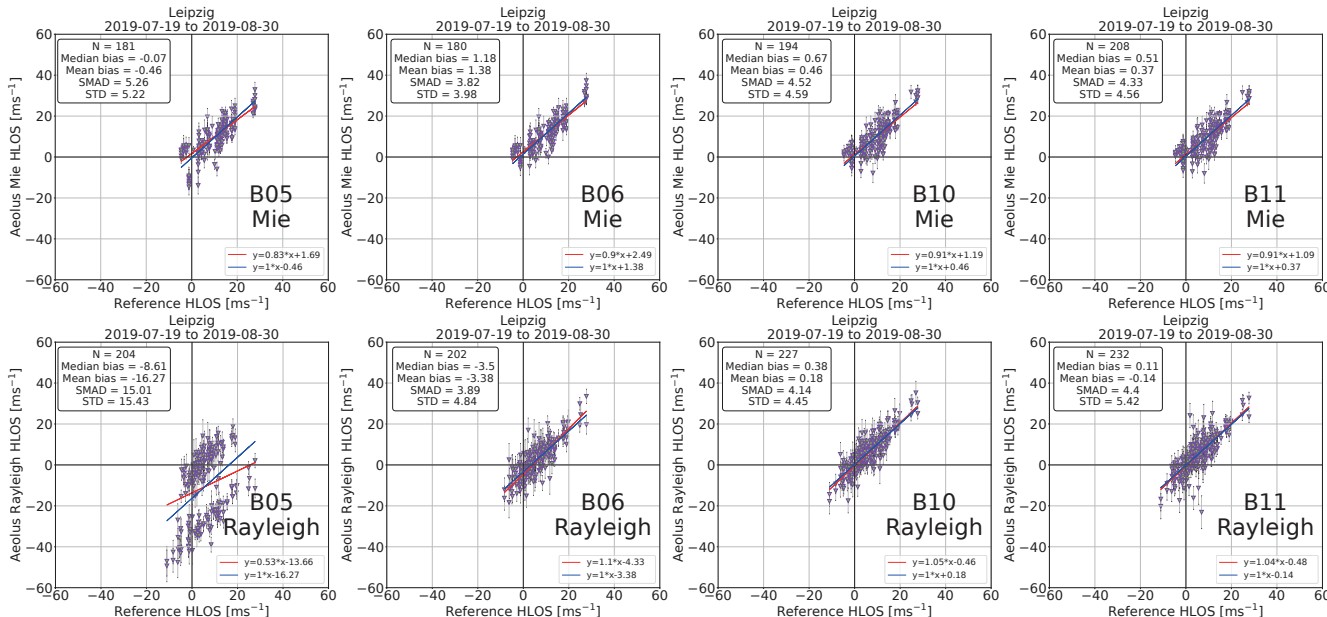

**Figure 10.** Aeolus performance as obtained by comparison to radiosonde launches at Leipzig (ascending orbit only) for Baselines 05, 06, 10, and 11 in the period from July to August 2019. Plots in analogy to Figures 6 and 7, left.

stronger negative HLOS winds of Aeolus compared to the radiosonde reference. This behaviour has completely vanished since Baseline 06.

### 6.1.2  B06 to B10 comparison

As shown in Fig 11, the improvement for B10 Rayleigh winds compared to B06 products becomes even more evident if one compares the longer period available for B06 and B10 only (see a dark yellow dashed rectangle in Fig. 1). Here, the longer time period from July to end of October 2019 could be considered covering 34 overpasses (17 for each orbit type – ascending and descending). Furthermore, the comparison focuses on the data quality depending on the orbit types, i.e., if Aeolus measured on an ascending or a descending orbit, to estimate the effect of the M1 temperature correction introduced since B09 (Weiler et al., 2021b). Therefore, we also did not include the Leipzig data (on ascending orbit only) here. There are obvious differences in performance of B06 between ascending and descending orbit type for the Rayleigh and Mie winds. E.g., a systematic error for the Rayleigh wind products of $-0.22 \, \mathrm{ms}^{-1}$ on the ascending orbit vs. $2.54 \, \mathrm{ms}^{-1}$ on the descending orbit. The majority of the outliers is seen for Mie winds on the descending orbit. These outliers in the Mie winds remain partly in Baseline 10, so that one can conclude that there must be other reasons for the discrepancy in the Mie wind measurement than the temperature deviation at the Aeolus telescope (Weiler et al., 2021b). For example, it might be atmospheric inhomogeneity which led to the result for which Aeolus measured about $-45 \, \mathrm{ms}^{-1}$, but the reference instrument only about $-30 \, \mathrm{ms}^{-1}$.

For B10, a systematic (random) error of 3.4 (6.2) and $-0.8$ (7.9) $\mathrm{ms}^{-1}$ was observed for the ascending and descending Rayleigh

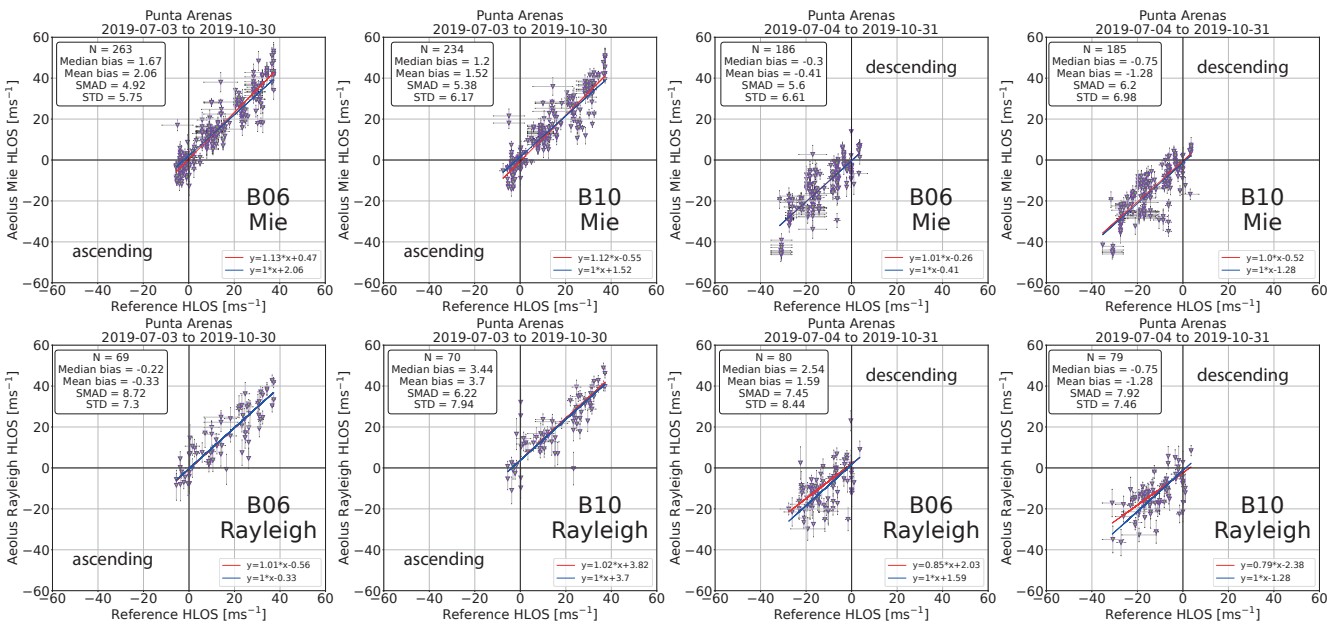

**Figure 11.** Comparison of Baseline 06 to Baseline 10 at Punta Arenas based on ground-based Doppler cloud radar observations separated for ascending and descending orbit type for the period of July to October 2019. Plots in analogy to Figures 6 and 7, left.

winds, respectively, indicating still a different behaviour between the different orbits types. The same is valid for the Mie winds, with systematic errors of about $1.2~\mathrm{ms^{-1}}$ (ascending) and $-0.75~\mathrm{ms^{-1}}$ (descending), while the random error is similar with around $6~\mathrm{ms^{-1}}$.

### 6.1.3 B07 to B11 comparison

We also analysed the difference between B07 and B11 (see dashed magenta rectangle in Fig. 1). The used data set covers Punta Arenas and Leipzig data from November 2019 to April 2020, thus southern hemispheric summer and northern hemispheric winter conditions. It is therefore well suited to statistically analyse the influence of implementation of the M1 telescope temperature correction on the data quality.

According to the results presented in Fig. 12, it becomes evident that between B07 and B11, like for B06 to B10, a significant improvement has to be attributed to the Rayleigh wind performance with much less outliers at both locations. Exemplary stated for Punta Arenas only (Fig. 12, left, bottom), a lower systematic ($-3$ vs. $-0.9~\mathrm{ms^{-1}}$) with almost equal random errors (about $7.8~\mathrm{ms^{-1}}$) were found for the Rayleigh winds. For the Mie wind performance at Punta Arenas (Fig. 12, left, top), the systematic error improved from $-0.6~\mathrm{ms^{-1}}$ to near 0, while the random error stayed equal with less than $5~\mathrm{ms^{-1}}$ with an even increasing number of available observations (295 to 393). The number of available observations of the Mie winds has also increased at Leipzig (Fig. 12, right, top), but with no significant changes in the errors, for both Mie and Rayleigh (Fig. 12, right) wind products.

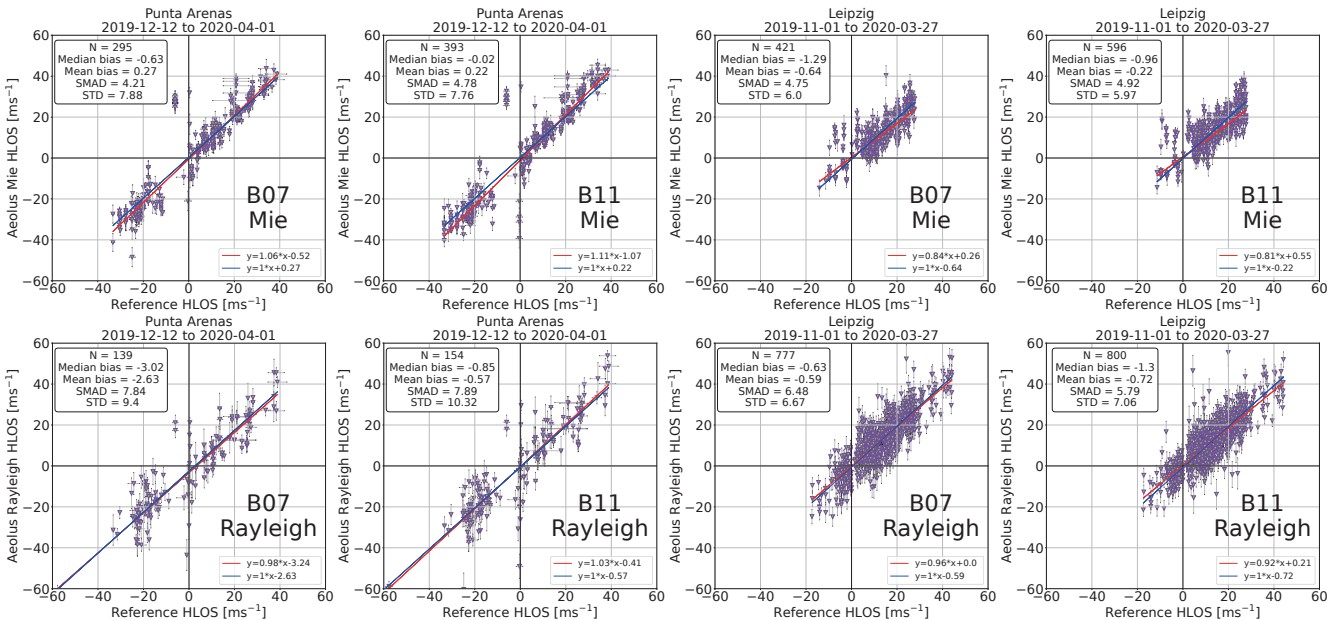

**Figure 12.** Aeolus performance at the reference stations of Punta Arenas (left, cloud radar, ascending and descending orbit) and Leipzig (right, radiosondes, ascending orbit) for Baselines 07 and 11 in the period from November 2019 to April 2020. Plots in analogy to Figures 6 and 7, left.

As the separation between the ascending and descending orbit is one key element for identifying the orbital bias effect, we separated the statistics according to that for Punta Arenas observations only and discarded the Leipzig data set (available for ascending orbit type only). The results are shown in Fig. 13. Without going into too much detail, we could not identify a significant improvement in the performance with respect to the Mie wind product in this specific data set covering 5 months of observations. Based on around 150 data points, the Mie systematic error was for B07 $+0.5 \mathrm{~ms}^{-1}$ on the ascending orbit while it was almost $-3 \mathrm{~ms}^{-1}$ on the descending orbit and thus remarkably different. With Baseline 11, however, the differences in the bias have even increased: $+1.5 \mathrm{~ms}^{-1}$ for the ascending orbit and $-3.6 \mathrm{~ms}^{-1}$ for the descending orbit, both for the Mie wind product. Thus, based on this limited data set, no improvement was found for the Mie winds. The Rayleigh winds, however, had an equal bias of around $-3 \mathrm{~ms}^{-1}$ for B07, which significantly decreased to $0.1 \mathrm{~ms}^{-1}$ and $-1.2 \mathrm{~ms}^{-1}$ (ascending and descending, respectively) for B11.

These comparisons discussed above are an indicator for the improvements made between different baselines - especially the importance of the implementation of the M1 temperature correction for the Rayleigh wind product. They are not meant as a general statement on the Aeolus performance as the analysed time period is short. A significantly large data set is key to determine statistically significant measures for a single validation site, like Leipzig or Punta Arenas. Therefore, the product performance is analysed between B10 and B11 on a longer time series in the following.

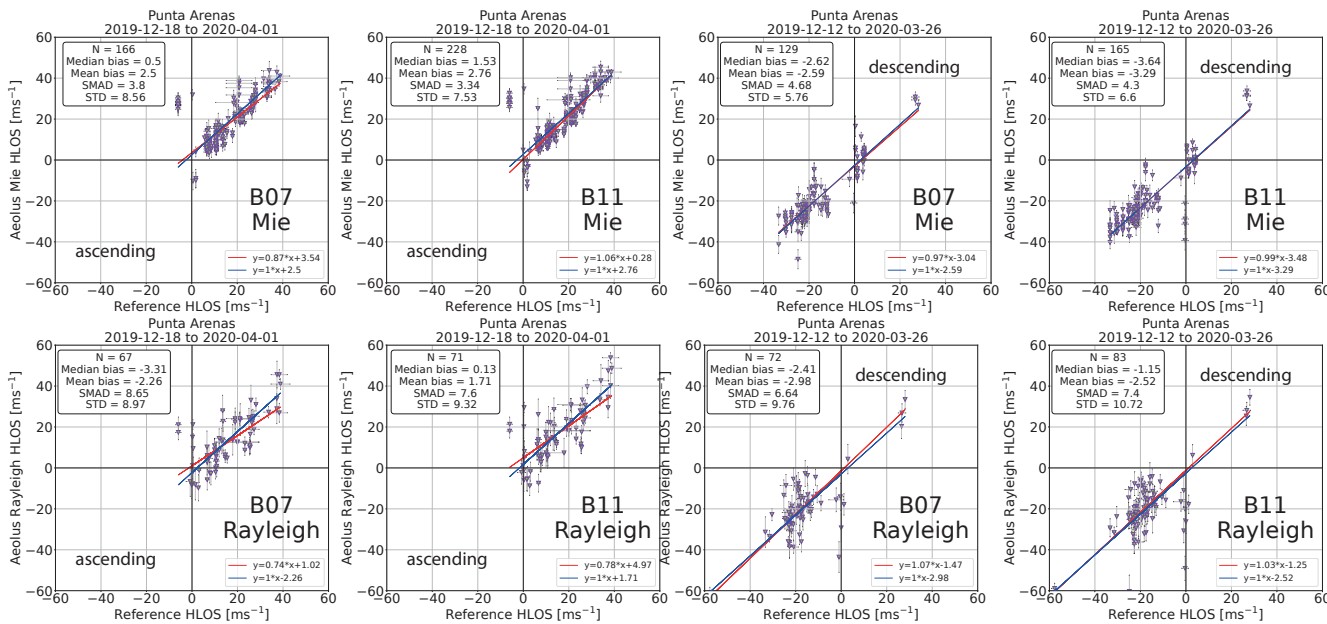

**Figure 13.** Comparison of Baseline 07 to Baseline 11 at Punta Arenas based on cloud radar observations separated for ascending and descending orbit type for the period from December 2019 to April 2020. Plots in analogy to Figures 6 and 7, left.

### 6.1.4 B10 to B11 comparison

As stated above, one major improvement step was reached by the introduction of the M1 telescope temperature correction with Baseline 09. Thus, it is also of interest to compare the algorithm versions beyond this Baseline. This is possible for B10 and B11, for which a significant amount of data are available in parallel as seen in Fig. 8 - light blue rectangles. The most important differences between B10 and B11 are the implementation of the Sat-LOS velocity correction, the reporting of the Rayleigh spot location and width values, and different signal-to-noise ratio (SNR) thresholds for classification of Mie and Rayleigh (ESA, private communication).

However, according to Fig. 14, left, no significant difference can be found at Punta Arenas between B10 and B11 for both Mie and Rayleigh wind products, despite the fact that about 5 % more Mie winds are available, which is most probably due to the new SNR thresholds for the wind type classification. In fact, the performance of the Rayleigh and Mie winds is slightly worse (overall small increase in systematic and random error). We can only speculate about the reasons for that, but it might be due to the new wind type classification or the newly implemented Satellite-LOS velocity correction. Similar findings are made for Leipzig, Fig. 14, right, for which the radiosondes could cover a much larger height range compared to the cloud radar observations in Punta Arenas, but only for the ascending orbit. Here, the absolute bias has slightly decreased from $-0.44$ to $-0.29\ \mathrm{ms}^{-1}$ for the Mie winds with similar random error, but more measurements for B11 as also observed for Punta Arenas.

For Rayleigh winds, no difference at all is seen, giving confidence that for this wind type Baseline 10 was already working well over the atmospheric range from 0 to 25 km - at least on Aeolus' ascending orbit over central Europe. If one separates the

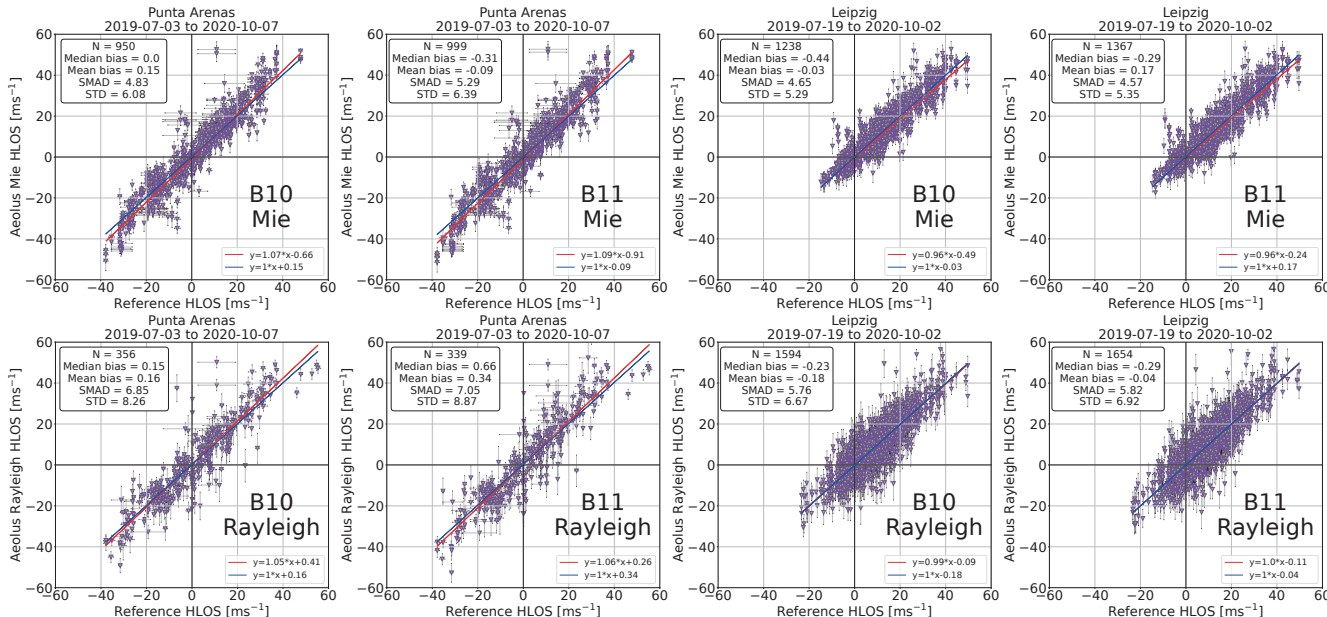

**Figure 14.** Aeolus performance at the reference stations of Punta Arenas (left, cloud radar, ascending and descending orbit) and Leipzig (right, radiosondes, ascending orbit only) for Baselines 10 and 11 in the period from July 2019 to October 2020. Plots in analogy to Figures 6 and 7, left.

orbit types for the statistical analysis, which is possible for Punta Arenas (Fig. 15), it is interesting to note that still a significant difference in the bias occurs between the two orbit types for both baselines. With respect to the comparison of the Mie winds on the ascending orbit between B10 to B11, the bias decreased, while on the descending orbit it increased (in terms of magnitude)

from $-1.7$ to $-2.4\ \mathrm{ms}^{-1}$. For Rayleigh winds, also like in Leipzig, no significant difference is seen at the geographic region of Punta Arenas between the two baselines, but with significant differences between the orbit types ($> 2\ \mathrm{ms}^{-1}$ vs. $\approx -2\ \mathrm{ms}^{-1}$). The random error remained equal between the baselines for both wind types.

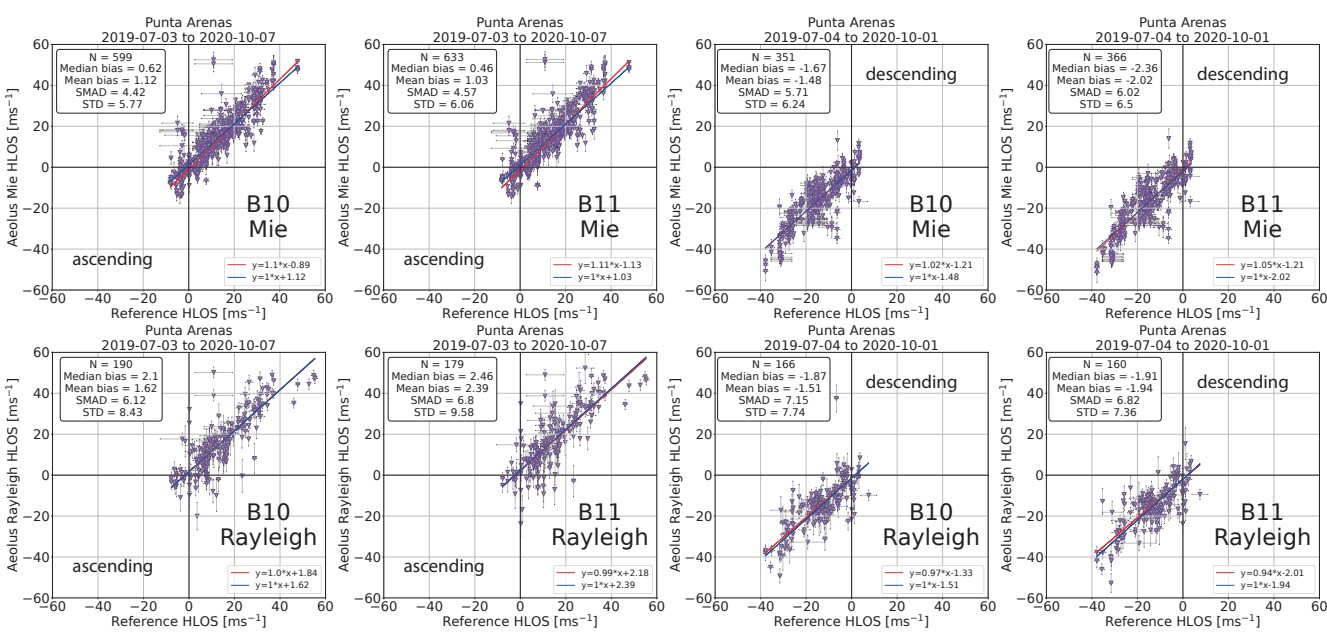

**Figure 15.** Comparison of Baseline 10 to Baseline 11 at Punta Arenas based on cloud radar observations separated for ascending and descending orbit type for the period from July 2019 to October 2020. Plots in analogy to Figures 6 and 7, left.

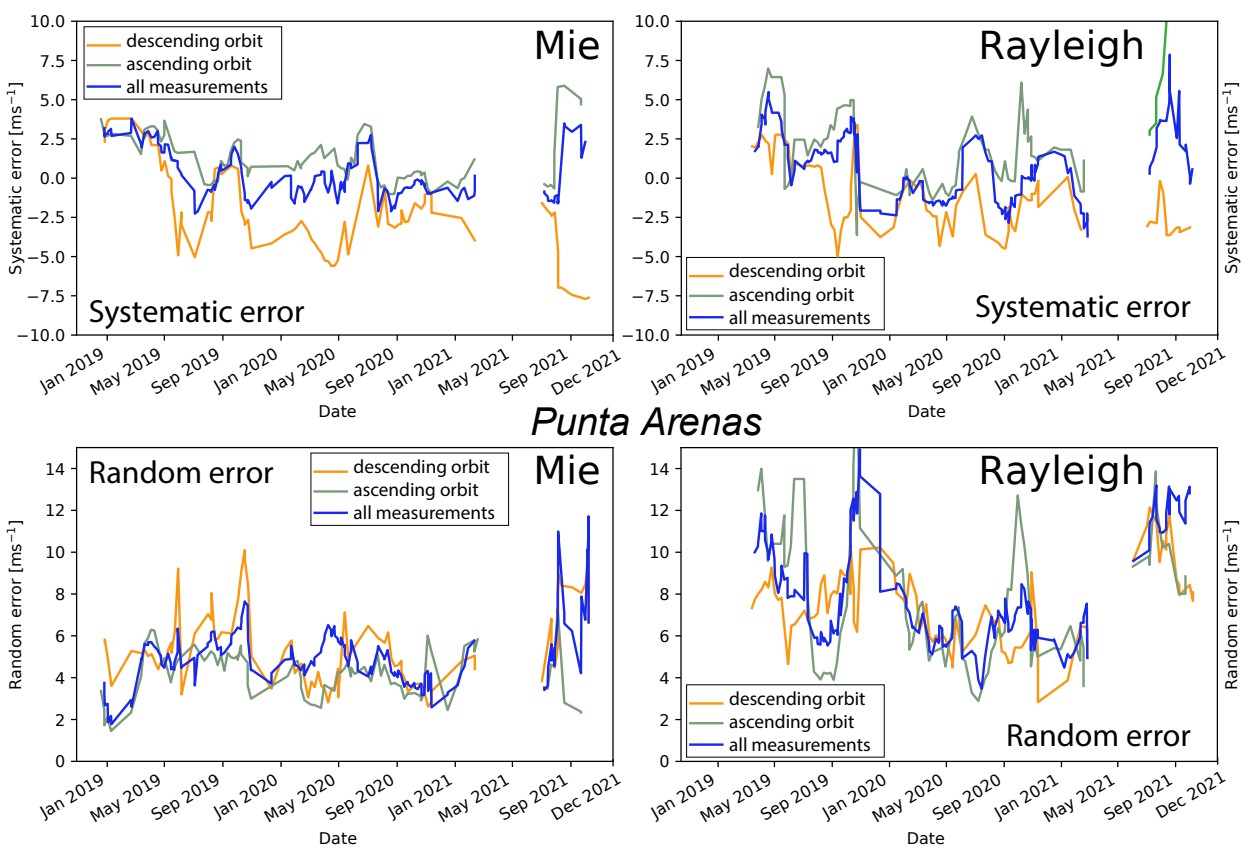

**Figure 16.** Long-term evolution of the derived systematic error (top) and random error (bottom) for the Aeolus Mie (left) and Rayleigh (right) wind products obtained at the Cal/Val station of Punta Arenas with cloud radar for all Aeolus observations (blue) and separated by orbit type (ascending: grey-green, descending: orange). A 50-days moving average window was applied.

## 6.2 Error evolution during lifetime

In the following, we assess the long-term performance of Aeolus. Thus, it is a mix of instrument performance and algorithm improvements. Figure 16 shows the temporal evolution of the systematic error (median bias, top) and random error (scaled MAD, bottom) for the Mie (left) and Rayleigh (right) products for the full 3-year data set at Punta Arenas. The temporal evolution was computed by using a 50-day moving average window, i.e., 7 full weeks with 14 Aeolus overpasses (7 for each orbit type). The newest baseline release was used for this analysis, i.e., for periods for which several baselines co-exist, the one with the highest number was analysed (i.e., B02: December 2018 until May 2019, B03: May 2019 until June 2019, B11: June 2019 –May 2021, B12: June 2021 – November 2021, B13: Dec 2021 – March 2022, B14: April 2022 – September 2022). The results are shown for all validation measurements (blue line) and are split into ascending (grey-green) and descending orbits (orange). Note that we here also present Aeolus data which is not yet public, i.e., from the very early mission time and thus this should not be regarded as final performance indicator of Aeolus. At the southern hemispheric mid-latitude location of Punta Arenas,

the systematic error of the Mie wind decreased from around 3 $\mathrm{ms^{-1}}$ in the beginning towards almost 0 $\mathrm{ms^{-1}}$ since May/June 2019 (for the combined observations including both orbit types, blue color). However, a difference between the wind products of the separate two orbit types (orange: descending, grey-green: ascending) becomes obvious especially for the period between October 2019 and August 2020. Sporadic outliers like in September 2020 might be due to certain weather conditions in Punta Arenas. The increase at the end of the observational period in 2021 might be attributed to the orbit shift performed for Aeolus in June 2021 and the resulting larger distances to the validation site. We also had to increase the radius from 100 to 120 km to still be able to validate both orbit types. Thus, the significant increase in magnitude of the systematic error might be attributed to the increased distance (mean distance 75 km and 85 km compared to 27 km and 75 km before the orbit shift). The random error of the Mie winds at Punta Arenas varies between 2 and 10 $\mathrm{ms^{-1}}$, but with most of the higher values after the orbit shift. The increase in random error since beginning 2019 might be attributed to the reduced return signal with laser FM-A and the calibration procedures after the laser switch (e.g., Parrinello et al., 2022). Additionally, the random error on the descending orbit shows a significant increase since the orbit shift.

For the Rayleigh winds, a significant improvement in terms of bias can be seen shortly after the start of the observations. Afterwards, the systematic error of the Rayleigh wind product seem to fluctuate between $-5$ and $+5$ $\mathrm{ms^{-1}}$ during the whole analyzed period, which might be an indicator for a reduced meaningfulness of the reference observation, which are available in cloudy atmospheric regions only compared to the Rayleigh winds available in clear air regions only, for a 50-day averaging window. Nevertheless, generally the systematic error for the descending orbit is mostly negative while the one for the ascending orbit is mainly positive. Therefore, the total retrieved bias fluctuates between positive and negative values. The random error of the Rayleigh winds has significantly improved in course of the mission lifetime from more than 10 $\mathrm{ms^{-1}}$ in the beginning of the observations to values of around 5 $\mathrm{ms^{-1}}$ in the middle of the analysed period. After the orbit shift in June 2021, the magnitude of the systematic and random error increased for all orbit types. Detailed reasons are yet unclear but might be simply attributed to the larger distances between Aeolus and the ground-reference instruments after the orbit change. The decrease in random error before the orbit shift is in contrast to expectations due to the parallel decrease of the atmospheric return signal of Aeolus and to other validation studies (e.g., Martin et al., 2021; Bley et al., 2022; Ratynski et al., 2023). However, published results of systematic and random error trends of Aeolus wind products are rare and to our knowledge, the only one trend series in the southern hemispheric mid-latitudes is provided in this study. Thus, the opposite than-expected trend might but not need to be attributed to atmospheric conditions.

A similar analysis was made for the Leipzig data set, which covers the ascending orbit only, but therefore is available until beginning of autumn 2022 (and thus includes Baseline 13 and 14, completely). The results are shown in Fig. 17. Similarly to Punta Arenas, the temporal evolution of the systematic and random error of the Mie and Rayleigh products for Leipzig have been analyzed by means of the median bias and the scaled MAD, respectively. A 50-day moving average window was applied, i.e., one smoothing window contained 7 overpasses, as the ascending orbit is analysed only for Leipzig.

The analysis for this location reveals, in accordance with the analysis for Punta Arenas before the orbit shift, that the systematic error of the Mie wind product was close to 0 $\mathrm{ms^{-1}}$ for the entire FM-B period (from June 2019 until September 2022). The random error for the Mie products has been stable at values around 4 to 5 $\mathrm{ms^{-1}}$ until mid of 2021. Afterwards, it decreased

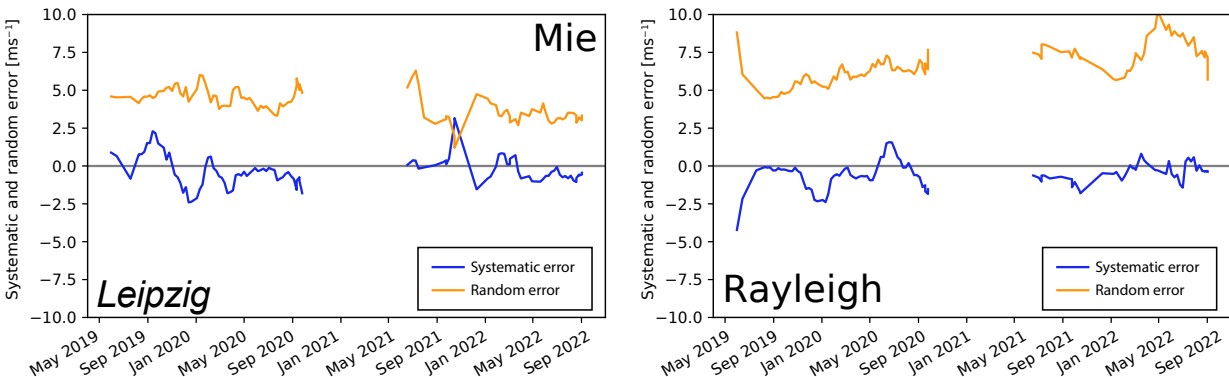

**Figure 17.** Temporal evolution of the derived systematic error (blue - median bias) and random error (orange - scaled MAD) for the Aeolus Mie (left) and Rayleigh (right) wind products obtained at the Cal/Val station of Leipzig (ascending orbit) with radiosondes. A 50-days moving average window was applied.

significantly followed by an increase and stabilization at 3 to 4 $\mathrm{ms}^{-1}$ until the end of the analysis period. Please note the obser-
vational gap which occurred during winter/spring 2020/2021 due to COVID-19 restrictions. Thus, no radiosonde (reference) data were available.

For the Rayleigh winds, also a positive performance trend was observed. The magnitude of the systematic error decreased significantly from values of around 4 $\mathrm{ms}^{-1}$ to magnitude values below 2 $\mathrm{ms}^{-1}$. The random error also decreased until the end of 2019 due to performance improvements obtained with the switch to laser FM-B but later mainly increased as a result of the decreasing return signal at Aeolus (e.g., Parrinello et al., 2022). The Rayleigh random errors at Leipzig, however, stayed always below 10 $\mathrm{ms}^{-1}$. Partially, a decrease in random error and thus an increase in performance was found as for example around January 2022, mainly caused by short phases of increasing return signal due to laser energy and alignment improvements. Similar overall trends in terms of random error as for the northern hemispheric mid-latitude station of Leipzig were reported by Martin et al. (2021)(northern hemisphere from 23.5°N to 65°N) and Bley et al. (2022)(tropical stratosphere).

### 6.3 Validation summary

We performed a validation analysis for both Aeolus wind products (Mie and Rayleigh winds) for the period for which our reference observations (Doppler cloud radar and radiosonde) were available. We thus considered several different baselines (see Table 1). The main results in terms of systematic and random error for Punta Arenas (Doppler radar) and Leipzig (radiosonde) are summarized in Table 3.

According to Table 3, left half, the systematic error of the Aeolus wind products could be significantly lowered with the changes introduced into the processing chain (different baselines). While in the early mission phase, systematic errors of more than 2 $\mathrm{ms}^{-1}$ (absolute values) were observed for both wind types, these biases could be reduced with the algorithm improvements, such as new calibration procedures or the M1 temperature correction with Baseline 09. Hence, since Baseline

**Table 3.** Overview of the systematic error (median bias) and the random error (scaled MAD) for the different baselines of Aeolus derived with the reference measurements at Punta Arenas (cloud radar, ascending and descending orbit) and Leipzig (radiosondes, ascending orbit). All values are in $\text{ms}^{-1}$.

| | Systematic error | | | | Random error | | | |
| --- | --- | --- | --- | --- | --- | --- | --- | --- |
| | *Punta Arenas* | | *Leipzig* | | *Punta Arenas* | | *Leipzig* | |
| | Rayleigh | Mie | Rayleigh | Mie | Rayleigh | Mie | Rayleigh | Mie |
| B02 | 3.1 | 2.73 | - | - | 10.97 | 4.68 | - | - |
| B05 | −8.18 | −3.09 | −8.61 | −0.77 | 19.82 | 6.87 | 15.01 | 5.26 |
| B06 | 1.51 | 0.91 | −3.14 | 1.09 | 7.98 | 5.57 | 4.76 | 4.23 |
| B07 | −3.02 | −0.63 | −0.63 | −1.29 | 7.84 | 4.21 | 6.48 | 4.75 |
| B10 | 0.15 | −0.0 | −0.23 | −0.44 | 6.85 | 4.83 | 5.76 | 4.65 |
| B11 | −0.0 | −0.41 | −0.46 | −0.35 | 6.79 | 5.05 | 5.77 | 4.59 |
| B12 | 0.56 | −0.69 | −0.86 | 0.42 | 11.2 | 5.4 | 7.33 | 4.26 |
| B13 | - | - | −0.37 | −0.04 | - | - | 5.97 | 4.24 |
| B14 | - | - | −0.34 | −0.67 | - | - | 8.49 | 3.2 |

10, a significant improvement of the Aeolus data was found leading to a low bias (close to $0~\text{ms}^{-1}$) for the Rayleigh winds and nearly similar values for the mid-latitudinal sites on both hemispheres. The systematic error of the Mie winds was already significantly reduced with Baseline 06. The random errors for the wind products as shown in Table 3, right part, are first decreasing with increasing baseline, but later increasing again as a result of the performance losses of the lidar instrument onboard Aeolus (Parrinello et al., 2022), but mainly only affecting the Rayleigh winds. The systematic error is only slightly affected by this issue, so one can conclude that the uncertainty introduced by the reduced atmospheric return signal received by Aeolus is mostly affecting the random error - of course on the cost of having less valid wind data, but at least no significant additional bias seems to be introduced.

# 7 Conclusions

To validate the novel wind lidar mission Aeolus, we have gathered long-term validation data at two mid-latitudinal sites but at different hemispheres. More specifically, we have performed regular radiosonde launches for the weekly Aeolus overpasses at Leipzig, Germany (51.35° N, 12.43° E), since May 2019. We also operated a scanning Doppler cloud radar in Punta Arenas, Chile (53.15° S, 70.91° W), so that horizontal wind speed and direction could be retrieved in the vicinity of clouds. We used all these data sources to validate the overall Aeolus performance with respect to mission time, the algorithm (baseline) version applied to Aeolus data, and the orbit type (ascending, descending, both). It was found that the deviation of the Aeolus HLOS winds from the ground-reference is of Gaussian shape. As systematic error indicator we thus applied the median bias of this distribution, while the random error was attributed to the scaled median absolute deviation in accordance to previous validation work on Aeolus and agreement within ESA and DISC (e.g., Lux et al., 2020a). It should be noted, that while the

radiosonde data performed at Leipzig on Aeolus' ascending orbit is covering the whole atmosphere from ground to ca. 22-25 km height, the Punta Arenas reference measurements with cloud radar are restricted to cloudy regions in the troposphere, i.e., the results from Punta Arenas are not representing stratospheric wind observations. Nevertheless, as clouds above Punta Arenas can be generally found at all altitudes and frequently, we consider the troposphere as well covered. The main findings, i.e., the systematic and random error by baseline, have been summarized in Table 3. In general, we have found an improving performance of the HLOS wind products with respect to the baseline development. This effect was however partly masked by the effect of lower instrumental performance of Aeolus during its lifetime, especially for the random error. From the whole Aeolus lifetime, we mainly analysed the period that was conducted with the spare laser called FM-B (starting with Baseline 05). Even when considering the issues with the emitted laser energy and the lower-than-expected received atmospheric return signal (e.g., Parrinello et al., 2022), which constantly decreased despite many efforts made, we can confirm the general validity of Aeolus observations during the lifetime. The systematic error of both wind products (Rayleigh and Mie) has significantly decreased as a result of newly introduced baselines with new calibrations and corrections. While at the beginning of the mission, absolute values as high as $5 \, \mathrm{ms}^{-1}$ were observed for the systematic error, it was continuously reduced to values close to $0 \, \mathrm{ms}^{-1}$ before the public release of the Aeolus data in April 2020. This proves the general concept of this Earth explorer mission to perform active wind observations from space. The random error has been indeed higher than requested by the mission requirements. But compared to the loss in return signal, the performance of Aeolus has been still in a range bringing a significant benefit for the numerical weather forecast as demonstrated, e.g., at ECMWF (Rennie et al., 2021), DWD (Martin et al., 2022), and NCMRWF (Rani et al., 2022). The data set gathered at Punta Arenas, Chile and Leipzig, Germany in the course of the validation project EVAA will stay of high value for the Aeolus mission. It can, for example, further be used to validate new algorithm versions applied to historical Aeolus data or to test new methodological approaches. Such efforts will continue even after the satellite has stopped measuring and will help to foster potential follow-on activities for active wind measurements from space as it is currently planned.

*Data availability.* The radiosonde data from Leipzig and the horizotnal wind data retrieved from the scanning Doppler cloud radar at punta Arneas are available at ESA Atmospheric Validation Data Centre (EVDC). Aeolus data are available via ESA Aeolus Online Dissemination System. The Cloudnet data used in this study are generated by the Aerosol, Clouds and Trace Gases Research Infrastructure (ACTRIS) and are available from the ACTRIS Data Centre using the following link:

https://hdl.handle.net/21.12132/1.c06a2c60c7504072.

*Author contributions.* HB has conceptualized the study and led the manuscript writing. JW has developed the algorithm for retrieving horizontal wind from ground-based Doppler radar (and lidar) under the supervision of MR and JB. EB has developed the methodology for the comparison of Aeolus to the ground-reference observations under supervision of HB. JW has finally incorporated these previous works for the long-term analysis under supervision of HB. BB and PS have been responsible for the Punta Arenas operations together with MR and

JB, HG for the radiosonde launches in Leipzig. UW contributed her expertise on space-borne profiling and general supervision. All authors have contributed to the intense discussions and the manuscript.

*Competing interests.* Ulla Wandinger is member of the editorial board of Atmospheric Measurement Techniques. Holger Baars is member of the Aeolus Science and data quality Advisory Group (Aeolus SAG). The authors have no further conflict of interest to declare.

*Disclaimer.* The presented work includes preliminary data (not fully calibrated/validated and not yet publicly released) of the Aeolus mission that is part of the European Space Agency (ESA) Earth Explorer Programme. This includes wind products from processor versions before Baseline 10 and/or aerosol and cloud products from before baseline version 11, which have not yet been reprocessed. The processor develop-
675 ment, improvement and product reprocessing preparation are performed by the Aeolus DISC (Data, Innovation and Science Cluster), which involves DLR, DoRIT, ECMWF, KNMI, CNRS, S&T, ABB, Serco and TROPOS, in close cooperation with the Aeolus PDGS (Payload Data Ground Segment). The analysis has been performed in the framework of the Aeolus Scientific Calibration and Validation Team (ACVT).

*Acknowledgements.* Many people are involved in performing the measurements which have been used in this Cal/Val study without whom such an extensive data set would not be possible. It's impossible to list all of them, but we honestly want to acknowledge all the work
that is done for installing instruments, keeping measurements running, launching radiosondes and analysing the data. This research has been supported by the German Federal Ministry for Economic Affairs and Energy (BMWi) (grant no. 50EE1721C). Furthermore, we also acknowledge the support through H2020-INFRADEV-2016-2017, Grant Agreement number: 7395302, and ACTRIS and the Finnish Meteorological Institute for providing the data set which is available for download from https://cloudnet.fmi.fi/. The measurements from Punta Arenas were produced by the Leibniz Institute for Tropospheric Research using resources provided by the Finnish Meteorological Institute and acquired in
the framework of the field experiment Dynamics, Aerosol, Clouds and Precipitation Observations in the Pristine Environment of the Southern Ocean (DACAPO-PESO), a research initiative from the Leibniz Institute for Tropospheric Research, Leipzig, Germany, in joint collaboration with the University of Magallanes, Punta Arenas, Chile, and the University of Leipzig, Leipzig, Germany. We also appreciate very much the fruitful discussions within the EVAA consortium (LMU, DWD, DLR) and with ESA.

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
