# Peer review of "Long-term validation of Aeolus L2B wind products at Punta Arenas, Chile and Leipzig, Germany"

_Atmospheric Measurement Techniques, 2022_

## Author Comment (AC1)

**Authors:**

We thank the reviewers for their careful reading and their very constructive criticism. We are confident that due to their comments the manuscript could significantly improve.

We addressed all issues brought up by both reviewers in the manuscript. A detailed response to the reviewer comments is found below in *"italic"*. Due to the amount of comments, the response is partly short, which does not mean that we didn't value the opinion of the reviewer, but rather to keep the discussion well-arranged.

Furthermore, we want to mention that during the revision of all statistical plots as requested by both reviewers, we realized that for the early baselines with the error unit in the Aeolus files being still m/s (before Baseline 8), Aeolus data with an error estimate higher than the stated thresholds were still included. We have removed those values now. The message remains the same, but some numbers have changed (improved). This has mainly affected Sec. 6.1.

Furthermore, we removed reference measurements (mainly radar) with an absolute error higher than 10 m/s these reference observations, so that a more meaningful comparison is possible. It has not changed much on the general statement, but some outliers (caused by the ground-based systems) are removed leading to a "better" systematic and random error statistic.

**Reviewer 1**

**General comments**

The results are correctly presented, and relevant conclusions are raised about Aeolus performance and the processing algorithms. However, the large number of plots makes the manuscript not easy to follow for the reader. Additionally, some figures should be reformatted in order to make them clearer to read and interpret.

*We reformatted all the figures, optimized the style and increased the font of the included text. We now hope that they are much better readable.*

**Specific comments**

The abstract is well written and structured. The concepts are well handled and a good overview of what has been done is provided. However, more precise information about the period used for each validation should be provided.

*Thanks, for the hint. We clarified more precisely now the analysed periods. To make the analysis more complete, we added all overpasses for baseline 14 in the Leipzig trend analysis shown in Fig. 17.*

Additionally, some concepts are not properly introduced (e.g., baselines, Mie cloudy, Rayleigh clear). On the other hand, some minor rephrasing could be performed to improve the understanding of the text.

*We tried to describe it better, however, having in mind that this is the abstract and not everything can be explained there.*

The introduction section is equally well written. The manuscript is well referenced and this review is significantly valuable. However, again the period considered in the validation activities is not totally clear, as it is specified when they start but not when the end.

*We have clarified this now in more detail.*

Additionally, the satellite is introduced in this section, but no information is given about the scene classification, quality flags, Aeolus errors, for example. This information is partially lacking also later in the text. More information about the satellite should be included in the introduction or in Section 2, together with the detailed description of the locations and the instrumentation.

*As the manuscript is submitted to the Aeolus special issue and many publications covering the validation of Aeolus are already within this special issue, we prefer to refer to other papers instead of describing everything again. But we have extended the description in Sec. 3 to give more details.*

On the other hand, this description is very well presented and detailed. In fact, Section 3.1 is overly detailed and it is not clear why this much details are needed for this specific study when so few details were given about Aeolus measuring technique and data processing.

*In our opinion, very detailed information on Aeolus is not necessary as shown in many other manuscripts in this special issue and other papers. On the other hand, the methodology for the retrieval of winds from Doppler cloud radar is rather new and the exact implementation (three methods) not published elsewhere. Thus, we have the feeling that some more details are needed. However, we tried to shorten this section a bit more and specify the reasoning in the introduction of this specific section.*

The methodology used for the validation is robust and has been widely tested in previous studies. However, more information about the overpasses is lacking (e.g., mean distance, impact of the orbit shift), which can be easily introduced. The presented case studies help to clarify the procedure followed. Nevertheless, the editing of Section 4 should be improved.

*Thanks for the hint, we have added the missing information in section 3 and also motivated the presentation of the case studies. Furthermore, we rephrased parts of Sec. 4 for a better understanding.*

Regarding Sections 5 and 6, mainly the editing of the figures should be improved. Some plots are hard to interpret, and the information is difficult to read, given the amount of tiny text included. All of the plots are interesting for the manuscript discussion. However, the authors should think of a better way of including that large number of plots. Additionally, given the number of plots, it is difficult to link the text with the figures while following the discussion.

*Thanks for the advice. We re-designed the figures the best way we could. We hope now that they are much better readable, even though we know that the amount of information in some figures is really high.*

On the other hand, there is not a clear criterion about when to use "Mie", "Mie cloud", "Rayleigh" or "Rayleigh clear". For example, "Mie cloudy" is used in the figures, while "Rayleigh" (and not "Rayleigh clear") is used, and while "Mie and Rayleigh winds" are used in the discussion. Some standardization should be performed or at least it should be made

clear. Nevertheless, the discussion of Section 5 and 6 are valuable and interesting results were raised.

*Thanks again for the statement. We standardized the wording throughout the complete text.*

**Technical corrections**

- Line 5. Specify when does the validation activities end. For Leipzig the orbit mode of the overpass considered was specified. But not for Punta Arenas.

*The radiosondes are still being launched, so no end date to be stated yet. But we stated the validation period now later in the text and added 4 more overpasses for the Leipzig comparison to complete baseline 14.*

- Line 12. Specify what is meant with "the era of the first laser". Indicate this period.

*Obsolete due to rephrasing.*

- Line 16. The concepts of "clear" and "cloudy" should be previously introduced.

*We think it is too extensive to explain these concepts in the abstract. But it is better explained now in the main text.*

- Line 22. This sentence should be rephrased.

*Done.*

- Line 25. What is meant with "concept" in this line? Please, rephrase.

*Done.*

- Line 37. Indicate in some way that "the technology" refers to Aeolus. Although Flament et al. (2021) and Siomos et al. (2021) provide an interesting address of Aeolus optical products at an earlier stage (presented in conferences), there are other relevant studies which entail a more in-depth study and validation of Aeolus optical products (published in the same Special Issue as the present work). Please, consider including the works of Abril-Gago et al. (2022) (https://doi.org/10.5194/acp-22-1425-2022) and Gkikas et al. (2022) (https://doi.org/10.5194/amt-2022-205), although the latter is still in review.

*Done!*

- Line 68. Indicate when the radiosondes stopped being launched. Also indicate that for Punta Arenas, ascending and descending node orbits were considered.

*The radiosondes are still being launched, so no end date to be stated yet. But we stated the validation period. Orbit notes are now also mentioned.*

- Line 71. Indicate when did the campaign end and whether the occasional radiosondes were considered in the validation or not. These radiosondes were not mentioned in the abstract.

*Done.*

- Line 84. Figure 1 is not referred anywhere in the manuscript. Please, consider mention it and include some discussion or remove it.

*Changed.*

- Line 93. Include also the radiosonde models here.

*Done!*

- Line 101. What is meant with "primarily". Were the radiosondes used or not?

*Rephrased. Thanks.*

- Line 114. Then, the Doppler cloud radar provides one horizontal wind profile each hour (of the PPI considered)? Provide more details about this. How was ensured the temporal collocation of this measurements with Aeolus products? At which time were they performed (i.e., every hour on the hour o'clock)? Discuss about the temporal representativeness of the measurements.

*The scans have been performed at minute 35 each hour and lasted for 60 seconds. This is now stated. The other issues are already discussed in Sec. 3.2. and Sec 4.1*

- Line 115. Discuss the reliability of the measurements for winds larger than 10.56 m/s, specially given one of the case studies provided.

*10.56 m/s are the radial velocity. The elevation angle was 85°. Thus, we can cover horizontal wind speeds up to 10.56 m/s / cos 85° = 121 m/s. It is expected that winds higher than this threshold did not appear in the troposphere.*

- Line 120. Give the information somewhere in Section 2.2.2 about when the radiosondes where launched (i.e., how much time before or after the overpass). A deeper discussion about the spatiotemporal collocation of the radiosondes is lacking in the text. Was the horizontal drift considered? Discuss about the representativeness of these measurements.

*We have stated now the launch time and added some discussion on the spatiotemporal colocation.*

- Line 131. Rephrase the information about the Lockheed Martin LMS6 radiosondes.

*Done*

- Line 133. The radiosondes where launched, but where they used? After reading the manuscript it is not clear at all. In several places it is stated that Aeolus products were validated with radiosondes in Punta Arenas, but no statistical results are given. Line 133 is a good place to make this clear.

*See comment above. We have not used them for statistics, but for case study validation also with respect to our cloud-radar-derived wind retrievals. We made this clearer in the text now.*

- Line 135. Please, consider rephrasing Section 3.1 to be more concise. A detailed description is given about how the Doppler cloud radar obtains the horizontal winds, but few or no information is given about how Aeolus or the radiosonde obtain these winds.

*See reply to the general comment above. We tried to more concise, but reviewer 2 asked for even more information. Therefore, the section was revised and we think we achieved a good compromise now. Furthermore, in section 3.2, a detailed description on the validation strategy is given.*

- Line 136. Please, rephrase. Mention TROPOS here gives the wrong idea that the instrument was placed in Leipzig for the activities.

*Thanks. We added "Punta Arenas" to make it clearer.*

- Line 180, subscript. A 100 km criteria was mainly used, except for Punta Arenas during the period after the orbit shift. Please, discuss this fact and how this could affect the representativeness of the winds captured by each instrument. Also mention and discuss the 100 km criteria recommended by ESA for Aeolus wind product validation activities.

*We added some further discussion with respect to the orbit shift and the therefore slightly increased radius. Furthermore, we added references for the choice of the 100 km radius threshold:*

- *Geiß, A., et al.: Methodology and Case Studies for the Validation of Aeolus Observations by means of Radar Wind Profilers, 2019.*
- *Cossu, F., et al.: Characterization of Aeolus wind measurement errors, 2022.*

- Line 182, Figure 3. Specify that the zoomed in products are the ones within the 100 km criteria. Also specify which classification they belong: cloudy or clear.

*We modified the plot so it should be clear now. Furthermore, we homogenised the wording so adding clear or cloudy is not necessary anymore.*

- Lines 182 and 183. Indicate the mean distance between the overpass and the station and their orbit mode. This information will provide an interesting discussion about the representativeness of the comparison results. Also discuss how does this mean distance and the overpass times change (or not) with time, and specially with the orbit shift. Two overpasses are mentioned for Leipzig, but only one is used. Mention this in the manuscript.

*See reply to general comment. We revised the respective section. Should be clearly stated now.*

- Line 184. Provide more details about the scene classification.

*See reply to general comment. After revising, it should be clear now. We prefer to refer to the already published description to not be too repetitive.*

- Line 185. Discuss if the 87 and 15 km horizontal resolution has been constant during the satellite mission. This could be linked to the new baseline releases.

*We refer now to Table 1 to make it clearer.*

- Line 192. Give more details about the "before the resolution was 87 km". This comment is related to the previous one.

*The date is stated there and together with the added information with respect to Table 1, we think it is sufficient information.*

- Line 194. Provide more details about the temporal collocation of the profiles. Additionally, this sentence should be rephrased.

*We have rephrased the text accordingly.*

- Line 200. This sentence needs rephrasing. Specify that the HLOS wind was averaged in height. Additionally, more information should be provided about Aeolus bins and bins thickness.

*Thanks for the advice. We have revised the sentences and added also references with respect to the Aeolus range bin settings:*

- *Stoffelen et al, 2005*
- *Straume et al, 2019*
  -
- Line 201, Table 1. Please, check if the start date for the operational processing of B12 is in May. Additionally, why were the subscripts not included in the "Additional info" column? If they are kept as subscripts, they should be numbered in some way.

*Checked! First data of B12 appear on the online dissemination centre for 26 May 2021. I guess you mean with subscript the footnotes and indeed it is a good idea, we change that and added it to the additional information of Tab.1.*

- Line 214. Type "Laser" with lowercase letters.

*Done*

- Line 219. Discuss here about ESA's recommended 100 km spatial criteria.

*This issue was now already earlier in the manuscript. We refer to that now.*

- Line 242. Please, rephrase.

*Done*

- Line 243. The profile was taken at 09:35, but it is not specified how much time it covers. Are these profiles always taken around half past? This temporal collocation of the Doppler cloud radar should be discussed before.

*After revising the methodology section, your question should be answered now. The PPI scans last 60 seconds and are performed always 35 minutes after the full hour. So, it is very close to the usual Aeolus overpasses (and this was the reason why the scan was performed at this time).*

- Line 244, Figure 4. The presentation of this figure should be improved. The arrow indicating the east direction of the middle plot is overlapped with a line and is barely visible. Additionally, the wind direction included in that plot is hard to interpret. In the left plot, the Doppler cloud radar can not be distinguished from the others. It should be specified which "Distance to lidar" the authors are referring to: distance to ground-track, distance to observations… In addition, it "lidar" should be replaced with "radar". It is not clear which time is given as "Ground-based instruments". The caption states "How are winds retrieves from the Doppler cloud radar", which is not really what Figure 4 presents. Furthermore, "Resulting winds retrieved with the Doppler cloud radar scans in regions cloudy occurrence" should be rephrased.

*We have revised the whole plot, so it should now be of much better quality addressing all the mentioned issues.*

- Line 248. Which frequency is that?

*You are right, the statement is misleading. We mean high spatial (vertical) resolution. We rephrased that.*

- Line 250. Please, rephrase.

*Done*

- Line 251. Some values should be provided in stead of only stating "excellent agreement". If GDAS data is going to be used, then some previous introduction should be given.

*We rephrased the sentence to make it more clear. GDAS data is only used on case-study-basis for consistency checks, which should be now made evident from the text.*

- Line 252. I recommend to change the tense of "have been" to "were".

*Done*

- Line 255. The discussion is very valuable. However, I would not make reference to the 87 km averaged Aeolus observation rather to the 100 km radius used, as that distance average is different for the Mie channel. Additionally, I would say that an optically thin cloud scenario could be also the case, so that Aeolus light could still penetrate.

*The whole paragraph was changed and the additional scenario mentioned. Thus, we hope it is now easier to follow our argumentation.*

- Line 262. It is not true for all the cases that the Rayleigh clear winds agree within the uncertainty range above 15 km.

*That's true, we rephrased an also discussed the potential occurrence of smoke at this altitude.*

- Line 264. Some observations are significantly lower than the GDAS, so the information is not totally true.

*True, information added.*

- Line 265. The following discussion gives the wrong idea that only cloud region could be analyzed. However, this is not true, as several radiosondes were launched and could be compared to both Aeolus Mie and Rayleigh winds. Then, it is worth mentioning here that the radiosondes could address the comparison in the other regions.

*We tried to state it clearer, in fact no radiosondes were used by us for the long-term validation at Punta Arenas.*

- Line 274. Information about the time when the overpass, Doppler cloud radar profile and radiosonde launch took place is lacking in Section 4.2.

*Information is given now*

- Line 277. Please, rephrase "considerable normal" or the whole sentence. Why using km/h now and not m/s as in the whole manuscript?

*Rephrased and units unified.*

- Line 280. Is there any evidence about the polar vortex shift over the region?

*Unfortunately, the weather maps we used at this time to characterize the scenario are not anymore available online. Thus, we skip this explanation as it is not the focus of this paper.*

- Line 285. I would recommend not to use "perfectly", especially because some points do not fit perfectly.

*Changed*

- Line 287. More information is lacking in the manuscript about how the radiosonde horizontal drift was solved, especially for the analysis in Leipzig. Please include this discussion in Section 3.2, for example. It could be interesting to include here a discussion with ESA's specifications of Aeolus performance under high wind conditions.

*In Leipzig, the radiosonde has been launched directly for Aeolus overpass as now described in Sec. 2.2.2, so that we considered that the 1--2-hour ascent of the radiosonde is representative for the atmospheric conditions during the Aeolus overpass and no systematic biases is introduced in the long-term validation due to the radiosonde drift. This is now stated in the text. The same accounts for the dedicated radiosondes in Punta Arenas, which were however, more irregularly launched.*
*With respect to high wind speeds, the mission requirement document of Aeolus states:*
*MR-90: The wind observation profiles shall have a dynamic range of +/-150 m/s along the HLOS direction*
*and*
*MR-95: The wind observation profile performances shall be applicable over a dynamic range of +/-100 m/s along the HLOS direction.*

*So, considering this fact, Aeolus should have been designed in a way to be able to observe such high wind speeds. We have no further information other than that. But if the reviewer has more information and is willing to share that, we are happy to include them in the manuscript.*

- Line 288. Give an approximate tropopause height here. Again, I would recommend not to use "perfectly".

*Rephrased – thanks! Tropopause was at 9.5 km.*

- Line 291. How can this be possible? Each Aeolus profile takes around 87 km, so how could two profiles fit in 100 km? It seems so simple that I might be wrong. I am especially concerned about the word "full" here.

*The validation radius is 100 km radius. Rayleigh winds are retrieved with 87 km horizontal resolution and coordinates are given for the start, end and the center of this 87-km observation. We used the center coordinates to check if the Aeolus profile is within the 100 km radius. Following that approach, usually 2 Rayleigh wind observations per profile are used for validation. Sometime it can be even 3 or only one, depending on the distance of the ground track to the ground location. We clarified that in the text in this line but also in Sec 3.2.*

- Line 298, Figure 5. The Doppler cloud radar profile is barely visible. Again, why using km/h and not m/s? Lidar should be replaced with radar as well.

*We redesigned this Figure and hope it is better visible now.*

- Line 298. I understand the limitation raised in this line. However, how can you be confident about it?

*Correct. We changed "confident" to "consider".*

- Line 314, Figure 6. A homogenization of the criteria used to name the channels is needed: use "Mie" or "Mie cloudy", but not both without any explanation before. Also rephrase (or not) "Rayleigh" accordingly.

  *Done*

  Additionally, including the error bars makes the plot harder to read. However, they could be kept, but further editing should be done to make the plots easier to interpret. Moreover, the units of the different coefficients and errors should be included. The same can be said to Figures 7, 9, 10, 11, 12, 13, 14 and 15. Especially Figures 9 to 15 need to be edited to make them easier for the reader to interpret.

*We have redesigned all Figures – see general comment above! We hope the readability of the plots have now improved a lot.*

- Line 313. Specify "Mie wind" and do not use "that wind". Additionally, discuss the comment about the slope and the mean bias.

*We added Mie wind now. Furthermore, we added: "as expected for a Gaussian distribution" to make the discussion with respect to slope and bias clearer.*

- Line 356, Figure 7. Replace "Ground Doppler" with Radiosonde" in the figure caption.

*Obsolete after Figure redesign.*

- Line 356, Table 2. Why was omitted the uncertainty of the Punta Arenas' slope?

*We simply forgot, thanks for spotting that!*

- Line 356, Figure 8. It should be described what the different rectangles mean. Additionally, a reference here to each of the following sections would be appreciated.

*Very good advice. We have done accordingly!*

- Line 358. In line 352 it was stated that the comparison with radiosondes (for Punta Arenas) was not worth it, thus it was not included. Why mentioning radiosondes here? In the following sections no results of the radiosonde comparison are given.

*Well spotted. Thanks! We removed it.*

- Lines 360 to 364. Although the information is more or less clear, this sentence should be rephrased to ease the understanding.

*Done!*

- Line 373, Figure 9. Using the same axis limits for all of the plots would help to compare the plots and interpret the results. At least, the axis should be symmetric, in order to make the 1:1 slope easier to observe. On the other hand, the font size is really small and most of the text in the plots is difficult to read. The same can be said for Figures 10, 11, 12, 13, 14 and 15.

*Redesigned – see general comment above. It really helped a lot. Thanks.*

- Line 375. Please, make reference to the square drawn in Figure 8 which corresponds to this period.

*Done*

- Line 379. Why is it obvious? Please, discuss. Also discuss why it indicates an improvement in the quality flags and error calculations.

*Ok, we state the numbers now and with the new plots it should be more obvious. We also discussed the quality flag issue now.*

- Line 393. Please, discuss why the Mie winds were already much more reliable for B05 and B06.

*According to Weiler, F et al, 2021, the impact of the telescope temperature variations were ten times less for the Mie winds than for the Rayleigh winds due to the instrumental nature of the two techniques. We discuss this now in the Manuscript.*

- Line 399. Specify when does this decrease took place?

*Done*

- Line 408. Please, make reference to the square drawn in Figure 8 which corresponds to this period.

*Done.*

- Line 412. It is not clear why the Leipzig data was not used here.

*Because we focus here on the ascending vs. descending orbit. As in Leipzig only the ascending orbit was available, we did not include it. We rephrased it and it should be clearer now.*

- Line 425. Please, make reference to the square drawn in Figure 8 which corresponds to this period. Also, rephrase the sentence, please.

*Done*

- Line 436. Why was Leipzig data discarded?

*Because we focus here on the ascending vs. descending orbit. As in Leipzig only the ascending orbit was available, we did not include it. We rephrased it and it should be clearer now.*

- Line 447. Please, rephrase.

*Done, thanks!*

- Line 458. Given the intercept value and the biases obtained, no significant differences were really obtained? Please, discuss.

*According to our understanding, the most significant change with Baseline 11 were*

1. *Sat-LOS velocity correction implemented*
2. *New wind type classification (copied from ESA confluence): "different SNR thresholds for classification of Mie and Rayleigh" and "and an option to transfer Mie SNR results to the Rayleigh channel was added",*

*Nevertheless, as discussed, we do not see a significant improvement but in fact a slight worsening. We stated this now more intensively and name some reasons, but we cannot make an in-depth discussion on the reasons due to the missing details.*

- Line 477. Is there any specific reason why a 28-days moving average was used?

*We intended to go for a 28-day smoothing window to ensure 4 weeks of overpasses (i.e. 8 overpasses in total and 4 for each orbit node). However, during the review phase we realized that the plot shown in the discussion paper was created with a 50-day smoothing averaged. This number was unfortunately hard-coded in the analysis code. We fixed this now, however the 28-day average which was then created was much noisier than the 50-day average (especially for Punta Arenas Rayleigh). Thus, we left this resolution window at 50 days which covers 7 weeks (14 overpasses) as a compromise. We state this now correctly in the text.*

- Line 478. "The most recent available baseline …" can be understood, but rephrasing would be appreciated.

*Done.*

- Line 504. Please, rephrase.

*Done.*

- Line 526. It would be really interesting if you discuss further about the fact that the Doppler could radar only captures the wind within clouds. How does this affect the representativeness of the results? Are they representative to all vertical regions of the atmosphere? You will not find much cloudy regions in the stratosphere, for example.

*Thanks. We discussed this now.*

- Line 530. Specify that the occasional radiosondes were launched in Punta Arenas. However, if these radiosondes were not used in the comparison, I would omit the information from the Conclusions.

*You are right, done!*

- Line 554. Please, make reference to the availability of the Doppler cloud radar and radiosonde data from Punta Arenas.

*We are still working on the upload of the radar data to EVDC which is more challenging than originally expected. If this does not work, we will publish the data at zenodo. If the reviewer likes, we can make the data available for him via an anonymous ftp account already now.*

- Line 695. The DOI link should be corrected.

*Done, thanks!*

---

## Author Comment (AC2)

**Authors:**

We thank the reviewers for their careful reading and their very constructive criticism. We are confident that due to their comments the manuscript could significantly improve.

We addressed all issues brought up by both reviewers in the manuscript. A detailed response to the comments of reviewer 2 is found below in *"italic"*.

Furthermore, we want to mention that during the revision of all statistical plots as requested by both reviewers, we realized that for the early baselines with the error unit in the Aeolus files being still m/s (before Baseline 8), Aeolus data with an error estimate higher than the stated thresholds were still included. We have removed those values now. The message remains the same, but some numbers have changed (improved). This has mainly affected Sec. 6.1.

Furthermore, we removed reference measurements (mainly radar) with an absolute error higher than 10 m/s these reference observations, so that a more meaningful comparison is possible. It has not changed much on the general statement, but some outliers (caused by the ground-based systems) are removed leading to a "better" systematic and random error statistic.

**Response to reviewer 2**

This manuscript presents the meaningful validation results for Aeolus via the long-term simultaneous measurements of ground-based instruments located at Leipzig (NH) and Punta Arenas (SH). It is impressive that the validation was performed for all product baselines (mainly from Baseline 05) after the switch-on of FM-B and the data products from different baselines are compared. Depending on mission time, baseline versions and the orbit types, the comparison results of this work are sufficiently presented and recognized appropriately.

The manuscript is well written and its contents are of high quality and scientific interest. The benefits of this study would be great for the accurate observations/high performances of Aeolus. Hence, I recommend the acceptance of this manuscript after the necessary revisions.

The specific comments are listed below:

1. Figure 1: This figure is not cited in this manuscript. This figure is somehow not necessary and the Aeolus Cal/Val stations are changed/updated worldwide. Thus, I recommend to remove this figure.

*Thanks for the hint, indeed the reference got lost during the working process. However, we still want to leave it as it might be not evident for every reader where Leipzig and Punta Arenas is located.*

2. The authors state that the Streamline Doppler lidars are applied in the supersites. Have the authors considered to include the simultaneous wind measurements from lidars into the comparison?

*Good point. We operated a Streamline Doppler lidar in Punta Arenas the whole campaign period while in Leipzig one instrument was available only a short time. However, due to the very low amount of particles in Punta Arenas, the performance of the Streamline was not optimum for the Aeolus validation. Mostly, wind retrievals were restricted to the local boundary layer. But due to the relatively long distance to the Aeolus ground track (partly*

*more than 50 km) and the complex orography, it was not useful to use this data for the Aeolus comparison. We state this now more clearly in the text.*

3. A 35 GHz Doppler cloud radar of type Metek MIRA35 at Punta Arenas, Chile was deployed and its data was used for the validation of Aeolus. In Section 2.2.1 of the manuscript (line 113 to line 115), you stated that "Once per hour, the stare mode (vertical profiling) was interrupted for a Range-Height-Indicator (RHI) and Plan Position Indicator (PPI, also called VAD - Variable Azimuth Display) scan. Only the PPI scans are considered for the horizontal wind retrieval.". Can I refer that the temporal resolution of the horizontal wind measured by the Doppler cloud radar is 1 hour? And how long does it take for one PPI scan, i.e., what is the accumulation time for each horizontal wind retrieval?

*We have revised the methodology section, to answer this question, which was also raised by reviewer 1. PPI scans lasted 60 seconds and were performed always 35 minutes after the full hour. So, it is very close to the usual Aeolus overpasses and has a resolution of 1 hour even though the measurement itself is made within 1 minute.*

Besides, the maximum unambiguous radial velocity and the resolution of the Doppler cloud radar were described in Section 2.2.1. Can you also provide the detection accuracy of this instrument? I think you can summarize the specifications and parameters of the instrument in table.

*Indeed, we could do this, but reviewer 1 even asked for less details on the ground-based instrumentation. Thus, we have to find a compromise with the level of detail for the used cloud radar. Therefore, we think it is better to cite Görsdorf et al for the general Mira 35 concept and state only the most relevant parameter for the TROPOS radar.*

1. In Section 3.1 of the manuscript, you described the horizontal wind retrieval method of the Doppler cloud radar and the three different fit methodologies are used to derive the horizontal wind vector in detail. But you just stated that "In the final data set, a best estimate is then computed which selects the method with the lowest error. This best estimate it then used for the comparison with the Aeolus winds." in line 176 to line 177. Can you provide the specific analysis result (e.g., errors of each method) and state which method was used? Is there one specific fit method for all the data set, or should the fit method be compared and chosen for every single wind profile retrieval?

*Thank you very much for the hint. The method which was finally used for the Aeolus validation depended on the atmospheric conditions. In most cases, the standard approach in line with Päschke et al. (2015) has been used. Rarely, one of the other fitting methods was set as best estimate, e.g., in case when Doppler folding occurred or the error with method 1 was higher as for the other methods. We extended the description in this section, so that it should be now better understandable:*

*"All three methods are performed for each range R to calculate the horizontal wind vector. In a final step, a best estimate is computed, which selects the method with the lowest error out of the three methods. In the data set, the retrieval results from all three methods plus the best estimate is stored.*

*This best estimate is then used for the comparison with the Aeolus winds, however not considering cloud-radar-derived HLOS winds with an error higher than 10 m/s."*

2. For the comparison criterion, why the temporal threshold of 1 hour is chosen? Theoretically, the ground-based radar performed continuous measurements of wind fields (except stare mode), hence there is no time difference between radar and Aeolus observations. For radiosonde-based comparison, assuming that we have a mean wind speed of 10 m/s for the horizontal wind and 100 km radius criterion, then it would take 100 km/10m/s=10000 s= ca. 2.8 h that the atmosphere moves above your ground site. Probably, the temporal threshold could be larger than 1 hour for radiosonde-referenced comparisons. In other words, the temporal threshold and spatial threshold criteria should be somehow consistent.

*Thanks for bringing this up. Indeed, it was probably confusing formulated. For the radiosondes, which have been launched dedicated to the Aeolus launch, no temporal restrictions are made. For the Cloud radar observations, however, we feel that the temporal threshold of 1 hour is needed as in Punta Arenas atmospheric conditions (because winds are available in clouds only) do change much faster than, e.g., the general advection pattern over Leipzig which is assumed to be completely recorded with the radiosonde. We clarified this now in the text*

3. Line 191 to line 194: the descriptions are "1–3 wind profiles fulfil this criterion of being within 100 km radius of the observational site (see green box in Fig. 3)" and "one can have up to 13–20 "Mie winds" for one altitude range around the 100 km of the ground-based location (see red box in Fig. 3).". But in the green box in Fig. 3, 7 wind profiles are presented, and I think there are more than 20 wind profiles in the red box. The wind profiles in the boxes are not all within 100 km radius of the observational site. You should mark the profiles fulfil the criterion further.

*Very great suggestion! We modified Fig. 3 accordingly and show only Aeolus observations within the 100 km radius.*

4. The legends in Fig. 9-Fig. 15 are unreadable. Alternatively, authors can enlarge them or list them in tables (like table 2 did) to help the readers to better understand the comparison results.

*We modified the plots accordingly. And hope they are now much better readable.*

The technical corrections:

1. The full stop at the end of the title should be omitted.

*Done.*

2. Line 1: …have been performed in Leipzig (51.12 N, 12.43 E), Germany, and at Punta Arenas (53.35 S, 70.88 W) should be have been performed in Leipzig (51.12 °N, 12.43 °E), Germany, and at Punta Arenas (53.35 °S, 70.88 °W). The degree signs are missing.

*Added! Thanks!*

3. Line 6: Does the "DACAPO-PESO" campaign have its full name? Please provide.

*Provided now in Sec 2.1.1.*

4. Line 181: According to your following description in line 182 to line 183, should it be two overpasses per week for each station?

*Changed.*

5. Line 192: "…vertical resolution" should be "… horizontal resolution…"

*Well spotted, thanks!*

6. CAL/VAL team (line 45) and CAL VAL teams (line 224) have to be consistent throughout the manuscript.

*Done.*

7. Figure 4: the Y-axis labels of the center and right panels should be consistent. Both should be "Height" or both be "Altitude".

*Thanks, the figure was completely revised and should now meet the requirements.*

8. Figure 6: the legend is hardly readable. Please polish it.

*Done, as for all other Figures which have been revised.*

9. The units must be written exponentially. Hence the "m/s" should be changed to "m s$^{-1}$". Please check this issue throughout the manuscript.

*Changed!*

10. Figure 16: the colors "pink" and "purple" sometimes are hardly to be distinguished. Please use some other colors.

*Changed! We tried our best to find distinguishable colours and hope that it is now better readable.*

---

## Author Response (AR2)

**Reply to proposed technical corrections:**

Technical suggestions: Normal font

*Our replies in italic.*
* * *
**Reviewer 1:**
*Many thanks for the careful reading!*

•Line 49. Replace "be" with "been".

*Done*

•Line 250. Replace "hear" with "here"

*Done*

•Line 270. Regarding former suggestion to Line 200, the authors replied that a reference to Stoffelen et al. (2005) and Straume et al (2019) was included. However, this does not seem to be finally included.

*Thanks for spotting, updated*

•Line 397, Figure 5. Still, the available bins for the Cloud Radar are not ease to see, as commented in former suggestion to Line 298. However, I understand the difficulties. It could be left unchanged. Additionally, due to current standardization, "clear" and "cloudy" could be omitted from the legend (also in Figure 4).

*Thanks, legends updated accordingly.*

•Line 422, Table 2. Why were the uncertainties of the slopes omitted from the table? It may be my mistake. In former suggestion to Line 356 I was asking you to include also the uncertainty for Punta Arenas, not to remove the rest.

*We added the slope uncertainties again, even though for the new data set they are all close to 0.*
* * *
**Reviewer 3**
*Thanks again for the careful reading!*

GENERAL COMMENTS:

1. Please restrict number of valid digits in the plots, text and tables

*We restricted now all numbers to a maximum of 2 digits throughout the manuscript.*

2. Please add the reference observation type in the fig. captions where missing (e.g. Doppler cloud radar)

*We updated all captions accordingly.*

SPECIFIC COMMENTS AND SUGGESTIONS:

- L10 products

*DONE*

- L24 The statement "…performance losses of the Aeolus emitter" is wrong for most mission phases and only true for the first laser – A phase. As laser – B performance was increased during the study time-frame, the sentence should be rephrased e.g. to performance losses of the lidar instrument.

*Changed.*

- L38 add "measured at 35° off nadir"

*Changed.*

- L81 remove "."

*Done!*

- *L200 numerically*

*Done!*

- L227 here

*Done!*

- L272 Other QC approaches have been shown to be required in e.g. Lux 2022b, especially for long term observations and different baselines, as the quality of the estimated error Aeolus product and its applicability as a QC-measure also changed. This should be mentioned in addition to what is stated in L390ff, and maybe stated to be considered for future validation of re-analysis products, to better show the product quality evolution and for better inter-validation study comparability. The error distribution in B05 Rayleigh comparison of Fig. 10 clearly shows that the purely estimated error - based QC is not enough, at least in this case.

*We added around line 272: "However, it needs to be mentioned, that for future validation studies of re-analysed wind products other quality control approaches should be considered as the quality of the estimated error will also change with changing baseline and thus also its applicability as an additional quality control parameter as, e.g., discussed in Lux et al. (2022b)."*

*and around line 450:*
*"The poorly estimated error product of the Aeolus Rayleigh winds at this baseline might have caused that invalid winds observations have been flagged as valid."*

- Fig. 4 Wind arrows in lower left plot are too small. Please increase and also describe the error bars in the right figure.

*We increase the size of the wind barbs and furthermore show only every second data point above 8 km for better visibility.*

- L293 Please also mention the Aeolus data content in Fig. 4 lower left

*We added "..indicated by the red rectangle in Fig. 4, left"*

- L325 This probably mis-classified Mie wind might also be interesting to be mentioned for a further future revisit in the context of re-analysis data validation. It might be an outlier.

*Thanks for the advice, we added: "Thus, the obviously misclassified Mie wind observation should be revisited in the context of the validation of future re-processed data"*

- L329 or L364 Please add a sentence on the horizontal homogeneity (HLOS wind spee variation) you expect from the ground instrument data variation statistics.

*In our opinion, an estimation of the influence of horizontal wind heterogeneity based on instrument variation statistics is not useful and could lead to wrong conclusions. We consider already the horizontal variation along the scanning cone of the cloud radar in the retrievals for the determination of the uncertainty. Using ground-based variation statistics for the estimation of horizontal heterogeneity would need to exclude many other effects like diurnal cycle, gravity wave activity, and so on, and thus open up a completely new aspect and room for speculations. We thus would like to stay with the uncertainty derived from the scans and leave the statements as they are.*

- Fig. 5 Please describe the error bars and increase the size of the color-bar and font of the insert.

*Done: "Radiosonde vertical error bars indicate the mean wind speed of the radiosonde averaged to the Aeolus range bins, Aeolus vertical error bars the respective range-bin extent."*

- L341 besides

*Done!*

- Fig. 6 and 7: Please state whether asc., desc. or both is shown and add Doppler cloud radar observations in the caption. In the B11 Mie linear regression result, correct + or − 1.07 (1.1) and increase font size for regression results. Please apply this also to Fig. 9 − 15 and correct +- occurrences. Describe hor. and vert. error bars in the fig. captions.

*We modified the Figures and numbers accordingly, but left the font size of the regression results as it is of minor importance and we do not want to even more overload the busy figures to not hide interesting results.*

- L387 use "sensitive" instead of "sensible"

*DONE!*

- L398 Geiß et al. 2022 is not available. Please update this reference and check the availability also for other references to this conference and add the web-site link

*That's true and a pity. Currently, we only can cite this reference without a link, but the ESA conference bureau was informed about the issue.*

- L416ff Better use "orbital" instead of "harmonic" bias

*Done and modified throughout the document.*

- In L 418 you mention that in Leipzig, only ascending orbits could be evaluated, so please indicated "ascending" also in Fig. 10 or its caption.

*Done as for all other Figure captions.*

- L422 ._A

*Done!*

- L444 Please rephrase to : The switch to laser FM-B had been already performed by this time

*Done!*

- L554 Please add the respective used baselines or indicate in Fig. 16 (and Fig. 17 accordingly)

*Done, we added int the text: "… number was analysed (i.e., B02: December 2018 until May 2019, B03: May 2019 until June 2019, B11: June 560 2019 –May 2021, B12: June 2021 – November 2021, B13: Dec 2021 – March 2022, B14: April 2022 – September 2022)."*

- L578 Other studies see an increase in Rayleigh random errors over time es expected from the atmospheric return signal decrease in the observed time frame (which could not be fully compensated by FM-B energy increases). Here, before the orbit shift, an improvement over time is observed, in contradiction to the expected increase from the lidar performance degradation and different to what is observed in Fig. 17 in Leipzig. It would be great, if you could expand one or two sentences on this in reference to other Aeolus validation studies.

*Thanks for the suggestion we added:*

*"The decrease in random error before the orbit shift is in contrast to expectations due to the parallel decrease of the atmospheric return signal of Aeolus and to other validation studies (e.g., Martin et al., 2021; Bley et al., 2022; Ratynski et al., 2023).*

*However, published results of systematic and random error trends of Aeolus wind products are rare and to our knowledge, the only one trend series in the southern hemispheric mid-latitudes is provided in this study. Thus, the opposite than-expected trend might but not need to be attributed to atmospheric conditions."*

*And:*

*"Similar overall trends in terms of random error as for the northern hemispheric mid-latitude station of Leipzig were reported by Martin et al. (2021)(northern hemisphere from 23.5°N to 65°N) and Bley et al. (2022)(tropical stratosphere)."*

- L593 Please replace "continuously" by "mainly" as there were a few phases with increasing signal due to laser energy and alignment improvements. This might be also worth mentioning related to the Rayl. rand. err. improvement seen in the Leipzig timeline around Jan 22 in Fig. 17 right.

*Done! And thanks for the hint, we added: "Partially, a decrease in random error and thus an increase in performance was observed as for example around January 2022, mainly caused by short phases of increasing return signal due to laser energy and alignment improvements. "*

- L607 Later Baselines cover the later FM-B period, where not the laser lost performance, but the send-return path. So please rephrase and e.g. replace "laser" with "atmospheric return signal" or "lidar instrument".

*Done!*

- L630 The return signal was always lower than expected. The laser energy was as expected or higher after improvements during the mission. You might want to correct your statement accordingly

*Done!*

[revised manuscript text omitted]